# Targeting ecto-5′-nucleotidase (CD73) and cAMP alleviates carotid body hyperactivity and reduces blood pressure in chronically hypoxic rats

Demitris Nathanael[1] , Andrew M. Coney[1], Dhaifallah Alotaibi[1,2], Prem Kumar[1] and Andrew P. Holmes[1,3]

[1]*Department of Biomedical Sciences, School of Infection, Inflammation and Immunology, University of Birmingham, Birmingham, UK*
[2]*Department of Respiratory Therapy, College of Applied Medical Sciences, University of Bisha, Bisha, Kingdom of Saudi Arabia*
[3]*Department of Cardiovascular Sciences, School of Medical Sciences, University of Birmingham, Birmingham, UK*

Handling Editors: Harold Schultz & Ken O'Halloran

The peer review history is available in the Supporting Information section of this article (https://doi.org/10.1113/JP289539#support-information-section).

**Abstract figure legend** CD73 promotes carotid body hyperactivity in chronically hypoxic animals. Exposure to chronic hypoxia (CH) ($F_iO_2$ of 12% for 10 days) increases the proportion of $CD73^+TH^+$ cells in the carotid body and induces basal chemoafferent hyperactivity. *Ex vivo* pharmacological inhibition of CD73 completely abolishes carotid body chemoafferent hyperactivity in CH. *In vivo* pharmacological inhibition of CD73 in the carotid body decreases arterial blood pressure in CH animals during normoxia but does not prevent a rise in ventilation when exposed to acute severe hypoxia.

**Demitris Nathanael** earned a bachelor's degree in biomedical science with a year in industry from the University of Birmingham. He gained early research experience at the University of Cambridge under Professor Randall Johnson and Dr Iosifina Foskolou, studying hypoxia's effects on immune cells. Returning to Birmingham, he was awarded a British Heart Foundation studentship to pursue a PhD in Dr Andrew Holmes' lab, investigating adenosine's role in carotid body hyperactivity during chronic hypoxia. He is currently a postdoctoral researcher at the Foundation for Liver Research, King's College London, exploring autonomic dysfunction in liver disease and its link to cognitive decline.

The Journal of Physiology

**Abstract** Chronic hypoxia (CH) is a key feature of chronic obstructive pulmonary disease (COPD), and carotid body hyperactivity is a major contributor to the cardiovascular pathology seen in these patients. CD73, an ecto-5′-nucleotidase that converts AMP to adenosine, is functionally important in the healthy carotid body. However it is unknown if CD73 contributes to carotid body hyperactivity associated with CH. The current work investigated whether selective targeting of CD73 could attenuate carotid body hyperactivity and ameliorate cardiovascular function in CH animals. CH rats ($F_IO_2$ of 12% for 10 days) exhibited basal carotid body chemoafferent hyperactivity, augmented hypoxic sensitivity and increased minute ventilation. Immunohistochemical analysis demonstrated that CH increased the number of tyrosine hydroxylase positive ($TH^+$) type I cells containing CD73 ($CD73^+TH^+$) within the carotid body. Pharmacological inhibition of CD73 using $\alpha, \beta$-methylene ADP (AOPCP) abolished CH-induced basal chemoafferent hyperactivity and normalised the heightened hypoxic sensitivity *ex vivo*. These effects were largely mimicked by the transmembrane adenylyl cyclase (tmAC) inhibitor SQ22536. *In vivo* administration of AOPCP to CH animals caused a dose-dependent decrease in arterial blood pressure, an effect that was attenuated following carotid sinus nerve section (CSNX). Therefore this research identifies CD73 and tmAC as important determinants of carotid body hyperactivity associated with CH. Effective targeting of CD73 alleviates carotid body hyperactivity, leading to a decrease in blood pressure, without abolishing reflex responses to hypoxia. These findings should be a first step towards the development of selective CD73/tmAC targeted therapies in the carotid body to safely improve cardiovascular outcomes in CH patients.

(Received 19 June 2025; accepted after revision 8 October 2025; first published online 16 November 2025)

**Corresponding authors** D. Nathanael and A. P. Holmes: Department of Biomedical Sciences, School of Infection, Inflammation and Immunology, University of Birmingham, Birmingham, UK, B15 2TT. Emails: DXN670@alumni.bham.ac.uk and a.p.holmes@bham.ac.uk

**Key points**

- Chronic hypoxia (CH), a key feature of chronic obstructive pulmonary disease, causes carotid body hyperactivity and cardiovascular pathology. The current work evaluated whether ecto-5′-nucleotidase (CD73) is important in promoting carotid body hyperactivity in CH animals.
- CH increases the proportion of type I cells containing CD73 ($CD73^+TH^+$) in the carotid body.
- Pharmacological targeting of CD73 and transmembrane adenylyl cyclases (tmAC) abolishes basal carotid body chemoafferent hyperactivity in CH and normalises the exaggerated responses to acute hypoxia *ex vivo*.
- *In vivo* inhibition of CD73 induces a dose-dependent decrease in arterial blood pressure in normoxia, which is attenuated following carotid sinus nerve section. However AOPCP does not abolish cardiovascular-respiratory responses to severe hypoxia.
- CD73 and tmAC are important in promoting carotid body hyperactivity in CH. Selective targeting of CD73 and cAMP signalling may safely reduce carotid body hyperactivity and thus improve cardiovascular outcomes in chronically hypoxic patients.

# Introduction

Carotid body hyperactivity is recognised as a key factor in the development and progression of cardiovascular pathology in multiple illnesses, including obstructive sleep apnoea, chronic heart failure, essential hypertension, diabetes, obesity, chronic inflammation and chronic obstructive pulmonary disease (COPD) (Abdala et al., 2012; Conde et al., 2023; Felippe et al., 2023; Incognito et al., 2025; Iturriaga, 2023; Phillips et al., 2018; Prabhakar et al., 2023; Schultz et al., 2015; Shin et al., 2019; Żera et al., 2025). In COPD patients carotid body hyperactivity contributes to the exaggerated sympathetic nerve activity and elevated arterial stiffness (Phillips et al., 2018; Stickland et al., 2016). Furthermore acute silencing of the carotid body using a combination of dopamine and hyperoxia significantly improves vascular conductance, reduces central pulse wave velocity and decreases blood

pressure in COPD patients (Phillips et al., 2018). Thus effective targeting of the carotid body may offer a new therapeutic approach to improve cardiovascular outcomes in this important and rapidly growing patient population (Boers et al., 2023). However at present there are no treatments used clinically that directly target the carotid body, highlighting a clear need for new device and/or drug development in this area.

A key feature of COPD is exposure to chronic hypoxia (CH). Respiratory adaptation to CH involves an increase in baseline ventilation and, in some cases, an augmented hypoxic ventilatory response (HVR) (Aaron & Powell, 1993; Bishop et al., 2013). This ventilatory acclimatisation is thought to be highly dependent on carotid body plasticity and hypersensitivity (Bishop & Ratcliffe, 2025; Powell, 2007). It has been proposed that both hypoxia inducible factor (HIF)-1$\alpha$ and HIF-2$\alpha$ are involved in the morphological and functional adaptation of the carotid body to CH (Cheng et al., 2020; Kline et al., 2002; Lam et al., 2008). Multiple studies have demonstrated an expansion in tyrosine hydroxylase positive (TH$^+$) type I cells in the carotid bodies in response to CH (Fielding et al., 2018; Pardal et al., 2007; Platero-Luengo et al., 2014; Sobrino et al., 2018; Wang et al., 1998). This coincides with modifications in gene expression, enhanced catecholamine synthesis and persistent rises in basal chemoafferent activity (Conde et al., 2012b; Gonzalez-Guerrero et al., 1993; Mosqueira & Iturriaga, 2019). However it remains unclear what underpins this chronic elevation in basal activity in CH and whether it can be reversed by pharmacological intervention. Furthermore in response to single levels of acute hypoxia some reports suggest an exaggerated rise in chemoafferent outflow in CH (He et al., 2005; He et al., 2010), whereas others do not (Conde et al., 2012b). What remains to be determined is if the single-fibre chemoafferent hypoxic sensitivity of the carotid body is altered by CH, that is, if there is a different PO$_2$ threshold at which it begins to respond to hypoxia, and if so, what is the underlying mechanism?

A potential pharmacological target within the carotid body is CD73, an ecto-5'-nucleotidase that converts AMP into adenosine. In the healthy rat carotid body CD73 protein is expressed in both type I and type II cells (Sacramento et al., 2019; Salman et al., 2017), where it functions to control adenosine concentrations in normoxia and hypoxia (Conde & Monteiro, 2004). We have previously shown that CD73 inhibition decreases basal chemoafferent activity and reduces hypoxic sensitivity in control carotid bodies (Holmes et al., 2015, 2018). This is possibly achieved through changes in adenosine generation acting via A$_{2A/2B}$ receptors coupled with increases in cAMP (Conde & Monteiro, 2004; Conde et al., 2006; Livermore & Nurse, 2013). Accordingly we have also reported that both adenosine receptor blockers and tmAC inhibitors decrease basal chemoafferent activity and hypoxic sensitivity in control carotid body preparations (Holmes et al., 2015, 2018). However the functional importance of CD73 and tmAC in mediating carotid body hyperactivity associated with CH is unknown. In addition it is unclear if *in vivo* inhibition of CD73 in the carotid bodies of CH animals modifies cardiovascular-respiratory parameters in normoxia and/or hypoxia, and if this is a safer approach compared to complete surgical bilateral carotid sinus nerve section (CSNX).

CD73 expression is known to be regulated by hypoxia, as evidenced in multiple different cell types (Alcedo et al., 2021; Sperber et al., 2023; Synnestvedt et al., 2002). Interestingly carotid body CD73 mRNA expression is reported to increase in response to CH (Salman et al., 2017). Thus we hypothesised that heightened CD73 expression and function may contribute to carotid body hyperactivity induced by CH. Therefore the aim of the current study was to evaluate whether targeting CD73 and/or tmAC attenuated carotid body hyperactivity in CH animals, leading to improvements in cardiovascular function.

## Methods

### Ethical approval

All experiments and procedures were performed in accordance with the UK Animals (Scientific Procedures) Act 1986 and were approved by the University of Birmingham Animal Welfare and Ethical Review Body and UK Home Office (Project licence PP9019875). All investigators understand the journal's policies regarding animal experiments, and adherence to these policies was maintained throughout this research. All work complies with the animal ethics checklist as outlined by the journal. Male Wistar rats aged 6–9 weeks, weighing 150–300 g ($n = 84$, Charles River Laboratories & Envigo, UK) were housed in temperature and humidity-controlled conditions (22 $\pm$ 1°C and 50 $\pm$ 10%) and 12 h-light/12 h-dark cycle (lights on at 0700). Animals had unrestricted access to food and water.

### CH rat model

Rats were housed in pairs within a normobaric environmental chamber (BioSpherix, USA), with unrestricted access to standard food and water. On the first day the F$_i$O$_2$ was gradually reduced from 21% to 12% over a 2 h period. The F$_i$O$_2$ was then maintained at 12% for 9–10 days. These animals are referred to as being chronically hypoxic, as they are exposed to chronic sustained hypoxia. Temperature and humidity were continuously monitored (Electronic Temperature

Instruments Ltd., UK), with silica gel used to absorb excess water vapour and soda lime to remove $CO_2$ and ammonia. Built-in circulating fans ensured constant airflow. Rats were monitored daily for welfare, including brief removal (10–15 min) from the chamber for routine husbandry tasks such as replacing silica gel and soda lime, changing the home cage, bedding and providing physical enrichment through handling. Control rats were kept in ambient air for the same duration and received the same enrichments. These animals are referred to as normoxic (N).

### Immunohistochemistry and confocal microscopy

Carotid bifurcations were surgically excised under 2%–5% inhalation isoflurane anaesthesia in $O_2$ at 1.5 L min$^{-1}$ delivered initially via an induction chamber and then via facemask. During the procedure surgical anaesthesia was verified by the loss of the pedal withdrawal reflex. Following the removal of the carotid bifurcations rats were culled via cervical dislocation, with death confirmed by exsanguination. Carotid bifurcations were immediately immersed in freshly made 4% paraformaldehyde in phosphate-buffered saline (PBS) (pH adjusted to 7.2) for 4 h at room temperature. The tissue was rinsed thrice in PBS and then placed into the dissecting chamber superfused with ice-cold PBS. The tissue was cleaned by removing any muscle, fat and nerves but leaving the superior cervical ganglion intact. The carotid body tissue was then placed into a 30% sucrose-PBS solution overnight at 4°C. The tissue was dabbed on paper towel to wipe away any excess sucrose and placed into a cryomold containing OCT and frozen using crushed dry ice and stored at −80°C until further use. Frozen carotid body tissue was sectioned at 10 μm using a cryostat (Bright Instruments, UK) set to −18°C and adhered onto charged microscope slides (Epredia Superfrost Plus, Fisher Scientific, UK). The sample was checked using a light microscope to ensure that the carotid body was present and its structural integrity intact. Slides were placed on a hot plate at 37°C for 2 h before being stored in tightly sealed boxes at −80°C. Slides were used within 24–48 h.

Frozen tissue slides were thawed at room temperature and rehydrated in PBS. Heat-mediated antigen retrieval was performed via immersion in citrate buffer (10 mM citric acid in diH$_2$O adjusted to pH 6.0 using NaOH solution and 0.05% Tween20) and incubated overnight in a 60°C water-bath. The following day the slides were allowed to cool to room temperature while still in the citrate buffer. After cooling slides were washed twice with PBS. Tissue sections were permeabilised with 0.3% Triton-X for 15 min and blocked with 1% BSA in 0.1% PBS-Tween20 (PBS-T) for 30 min at room temperature. A hydrophobic barrier was drawn around the sections using a PAP pen (ImmEdge, VectorLabs, UK). Primary antibodies targeting CD73 (rabbit polyclonal 1:100, ectonucleotidases.com, Canada, CAT #rNu-9L) (Fausther et al., 2012), tyrosine hydroxylase (mouse monoclonal 1:500, Cell Signalling Technology, UK, CAT #45648S) and PCAM-1 (CD31, goat polyclonal 1:500, Bio-techne, UK, CAT #AF3628) were diluted in 0.1% BSA in 0.1% PBS-T. Slides were placed in a humidified chamber and incubated overnight at 4°C. Slides were washed thrice with 0.1% PBS-T and incubated with fluorescently conjugated secondary antibodies (anti-rabbit Alexa Fluor 488 1:1000, CAT #A21206; anti-mouse Alexa Fluor 594 1:250, CAT #A21203 and anti-goat Alexa Fluor 647 Plus 1:1000, CAT #A32849, ThermoFisher, UK) for 1 h at room temperature in the dark. Following incubation slides were washed thrice with 0.1% PBS-T and given a final rinse in PBS. Vibrance anti-fade hard-set mounting medium containing DAPI (Vector Laboratories, H-1800, UK) was applied to the sections and cover slipped (13 mm round, #1.5 thickness, hydrolytic class 1 glass, VWR, UK). Slides were allowed to cure in the dark for 2 h at room temperature (exposed to air) and imaged using a confocal microscope within 48 h. Images were acquired using a Zeiss LSM 880 microscope (40× magnification, 1.2 NA, water objective) and a Leica Stellaris 8 microscope (40× magnification, 1.4 NA, oil objective). Both microscopes used a tile scanning function (10% stitching overlap), and all confocal settings were kept the same between N and CH samples.

### Image analysis and quantification

Due to the carotid body tissue being immersion-fixed, this resulted in residual red blood cells (RBCs), which exhibited significant autofluorescence (AF) due to the porphyrins in haemoglobin. The relatively high pixel intensity of RBC AF and complexity of the images made regular thresholding and the other conventional methods unsuitable for removing this false-positive signal. To address this a machine learning approach was employed using the Python-based software, *ilastik* (Berg et al., 2019). For training immunofluorescence images were manually annotated to label RBC pixels. This process was repeated across approximately four images to train the model. Once trained the software predicted pixel classifications and batch-processed subsequent images. The output was a segmented image with RBC pixels clearly distinguished. This segmented image was imported into Fiji, where RBC-specific pixels were isolated and converted into a binary mask. The binary mask was subtracted from the original raw image, effectively removing RBC AF. The mask could also be subtracted from other channels in the same image, ensuring accurate quantitative analysis of all proteins of interest. A step-by-step workflow for using

*ilastik* to remove RBC AF prior to image quantification is provided in Fig. 1*A–E*.

All quantification was performed in Fiji (version 2.14.0). Cell counts in each section were performed manually. The total area of positively stained signal was measured by first applying a threshold and removing noise (excluding outliers < 2 pixels). Threshold values were automatically determined by the software. However in cases of high background where automatic thresholding was unreliable, threshold values were manually set based on the approximate pixel intensity observed in the pre-immune serum condition (negative control). To quantify co-localisation/overlap Manders' coefficients were calculated using the JaCoP plugin in Fiji. These coefficients provide a measure of the proportion of one signal overlapping or co-localising with another in the same image (Dunn et al., 2011).

## Extracellular recordings of carotid body chemoafferent activity

Carotid bifurcations were surgically excised under 2%–5% inhalation isoflurane anaesthesia in $O_2$ at 1.5 L min$^{-1}$, delivered initially via an induction chamber and then via facemask. During the procedure surgical anaesthesia was verified by the loss of the pedal withdrawal reflex. Following the removal of the carotid bifurcations, rats were culled via cervical dislocation, with death confirmed by exsanguination. To preserve carotid body function and minimise severe hypoxic insult the excised tissue was immediately transferred to an ice-cold bicarbonate-buffered extracellular physiological salt solution containing (in mM) 25 NaHCO$_3$, 119 NaCl, 4.5 KCl, 1.2 MgSO$_4$.7H$_2$O, 1.2 NaH$_2$PO$_4$, 11 D-glucose and 2.4 CaCl$_2$, equilibrated with 95% $O_2$ and 5% $CO_2$. Carotid bifurcations were finely dissected to

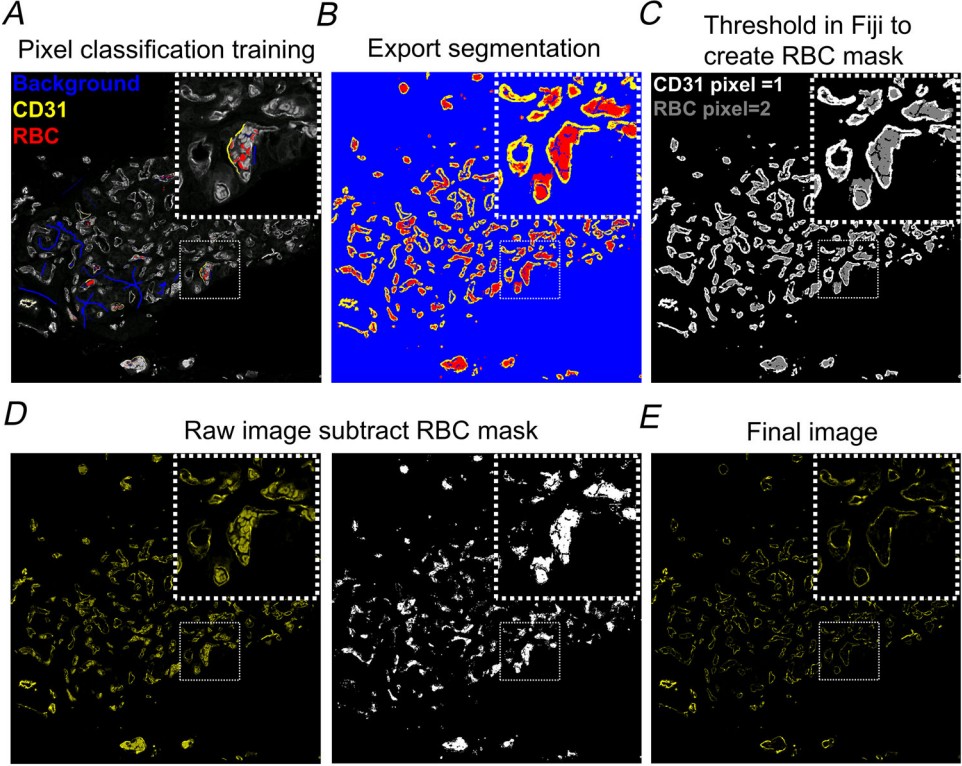

**Figure 1. Machine learning approach for detection and removal of red blood cell autofluorescence in carotid body sections**

*A*, a raw confocal immunofluorescence image of a carotid body section showing CD31$^+$ staining and red blood cell (RBC) autofluorescence (AF). Using the software *ilastik* different signals can be classified based on their pixel properties and can be labelled accordingly, background (blue), CD31 (yellow) and RBCs (red). Training was performed on four different sections, and the software could accurately predict the correct label for each pixel. *B*, a segmented image after pixels have been classified into their labels, which was exported into Fiji. *C*, segmented image in Fiji showing CD31$^+$ pixels (grey scale pixel value = 1) and RBC pixels (grey scale pixel value = 2). A threshold can be applied with a grey scale minimum/maximum of 2 to only display RBCs. Applying this threshold created a binary mask (bottom row, centre) which was subtracted from the original raw image *D*, *E*, final image containing CD31$^+$ signal, free from RBC AF, ready for analysis. Insets (dotted boxes) provide a closer view of the boxed regions.

further isolate the carotid body and carotid sinus nerve, which was sectioned to expose nerve fibres. To facilitate extracellular neuronal recordings the whole tissue was partially digested by incubation in a bicarbonate-buffered physiological salt solution containing 0.075 mg ml$^{-1}$ Collagenase Type II (Sigma-Merk, UK, CAT #C6885) and 0.0025 mg ml$^{-1}$ Dispase Type I (Sigma-Merk, UK, CAT #D4818), equilibrated with 95% $O_2$ and 5% $CO_2$ at 37°C for approximately 25–30 min.

Extracellular recordings of carotid body sensory activity were made from the cut end of the carotid sinus nerve, as previously described (Holmes et al., 2014; Pepper et al., 1995). The superfusate $PO_2$ was continuously measured throughout experiments using an $O_2$ electrode (ISO-OXY-2, World Precision Instruments, UK) placed immediately before entry into the recording chamber. The voltage from $PO_2$ and chemoafferent signals were digitised at 200 and 15 000 Hz, respectively, by a CED micro1401 (Cambridge Electronic Design, UK) and visualised on a PC running Spike 2 (version 7.2). The chemoafferent voltage signal was amplified by a total factor of 5000 and band-pass filtered between 50 and 20 000 Hz. During experiments the carotid body tissue was continuously superfused with the same physiological salt solution mentioned earlier. Basal chemoafferent activity was measured in a $PO_2$ of 300 mmHg and $PCO_2$ of 40 mmHg. This $PO_2$ was considered normoxic, as this level has been shown to generate a basal single-fibre chemoafferent discharge frequency within the range of 0.25–1.50 Hz (Holmes et al., 2014) in this *in vitro*, superfused preparation. This is consistent with the observed *in vivo* single-fibre chemoafferent frequency range recorded in healthy rats during arterial normoxia (Vidruk et al., 2001).

Initial experiments aimed to characterise the pharmacology of a CD73 inhibitor, $\alpha,\beta$-methylene ADP (AOPCP, Bio-techne, UK, CAT #3633) and transmembrane adenylyl cyclase inhibitor, SQ22536 (Bio-techne, UK, CAT#1435). Cumulative concentration-response experiments were performed in normoxia in both N and CH carotid bodies. Concentrations were given for 5 min, and discharge frequencies were taken as the average of the last 60 s. After drug washout and returning chemoafferent activity back to baseline levels, hyperoxia was given as a positive control to achieve maximum reduction in chemoafferent activity. To determine adenosine sensitivity 100 μM adenosine was added to the superfusate for 5 min, a concentration known to give maximal effect (Vandier et al., 1999).

To induce hypoxia the superfusate $PO_2$ was gradually ramped down from 300 mmHg to approximately 50 mmHg at a constant $PCO_2$ of 40 mmHg (using high precision variable flow meters, Cole-Parmer, UK), until the discharge reached 10–20 Hz before rapidly switching to a hyperoxic solution (95% $O_2$, 5% $CO_2$). Hypoxic responses were repeated in the presence of AOPCP or SQ22536 and again after drug washout. Electrical activity originating from single fibres was used for analysis. To investigate the impact of AOPCP and SQ22536 on graded hypoxic responses, the single-fibre chemoafferent frequency was plotted against the superfusate $PO_2$, and fitted to an exponential decay curve with offset:

$$y = a + be^{-cx},$$

where, $y$ is the single-fibre frequency in Hz, $x$ is the $PO_2$ (in mmHg), $a$ is the discharge frequency as $PO_2$ tends to infinity (offset), $b$ is the theoretical frequency when the $PO_2$ is 0 mmHg (minus the offset) and $c$ is the exponential rate constant. To evaluate changes in the set-point or $PO_2$ threshold required for response initiation, it was necessary to determine if the chemoafferent-hypoxic response curve was left or right shifted. To do this for a given discharge frequency the corresponding $PO_2$ could be calculated using the inverse function of the exponential decay curve:

$$x = - \left( \mathrm{Ln} \left[ (y - a) / b \right] \right) / c,$$

where $x$ is the $PO_2$ (in mmHg), $y$ is the single-fibre frequency in Hz and $a$, $b$ and $c$ are constants. The required $PO_2$ at a frequency of 5 Hz was used to quantify a $PO_2$ shift because this frequency lies on the exponential region of the hypoxic response curve but is not of a magnitude at which the discharge has peaked and/or started to diminish (Holmes et al., 2016, 2018).

To reduce animal numbers both carotid bodies were used from a single animal for chemoafferent recordings. However a single carotid body was only exposed to one of the drug agents, either AOPCP, SQ22536 or adenosine.

## Whole-body plethysmography

Baseline ventilation and hypoxic responses were assessed in freely moving, conscious rats using whole-body plethysmography (EMKA Technologies, France) (Alzahrani et al., 2021). Respiratory signals were acquired using iox2 (EMKA Technologies, version 2.10.8), converted to a European Data Format and analysed using LabChart 8 (ADInstruments). Minute ventilation ($V_E$) was calculated as tidal volume multiplied by respiratory frequency. All rats were weighed prior to experiments; therefore, tidal volume was normalised to body weight, and thus $V_E$ was expressed as ml min$^{-1}$ g$^{-1}$. High precision mass flow controllers were used to set a constant gas flow rate of 2 l min$^{-1}$ for normoxic/hypoxic gas mixtures and quantified using a gas analyser (ADInstruments, UK). Rats were left to acclimatise to the plethysmography chamber for a minimum of 20 min in normoxia before recording any data. Baseline ventilation was measured in an $F_iO_2$ of 21% for N ($n = 4$), and 12% for CH

($n = 12$). These values are defined as normoxia for each group. Hypoxia was achieved by lowering the $F_iO_2$ to 8% (balanced nitrogen) for 10 min before returning to normoxia. An average of the last 10 s of every minute was used to plot the respiratory time courses. The HVR was calculated as the difference between the mean of the last 3 min of normoxia and the mean of the last 3 min of hypoxia (steady state). Ventilatory roll off during hypoxia was calculated as the percentage reduction from the absolute peak during hypoxia to the last minute of hypoxia. Following experimentation animals were either culled by exposure to carbon dioxide gas in a rising concentration, death confirmed by cervical dislocation or placed back in their home cage at 12% $F_iO_2$ for CH and ambient air for N. In the latter instance animals subsequently had carotid bodies isolated 1 day later for analysis of chemoafferent activity or CD73 protein using immunohistochemistry.

### *In vivo* measurements of carotid body-mediated cardiovascular and respiratory function

Rats were initially anaesthetised with 2%–5% inhalation isoflurane anaesthesia (delivered via induction chamber and then facemask) in $O_2$ at 1.5 L min$^{-1}$, and then the right femoral vein was cannulated. Isoflurane was removed, and anaesthesia was maintained with alfaxalone (Alfaxan, Vetoquinol Ltd., UK) infusion at a rate of 7–12 mg kg$^{-1}$h$^{-1}$ delivered intravenously (i.v.). Surgical anaesthesia was monitored and verified by the loss of the pedal withdrawal reflex; 0.1 ml boluses of alfaxalone were given, as necessary. We have previously shown that the use of alfaxalone preserves carotid body-mediated cardiovascular-respiratory reflex responses to multiple different stimuli, including hypoxia, hypercapnia and hypoglycaemia (Alzahrani et al., 2021; Thompson et al., 2016). The right femoral artery was cannulated and connected to a pressure transducer (ADInstruments, UK) to record arterial blood pressure (ABP) from which heart rate was also derived. The left femoral vein was cannulated for bolus injections of AOPCP. The trachea was cannulated and connected to a spirometer (ADInstruments) to measure airflow in which $V_E$ was derived from tidal volume and respiratory frequency. In all animals superficial glands and muscles of the neck were retracted or removed to expose the carotid bifurcation, glossopharyngeal nerve and carotid sinus nerve before measurements started. This allowed for rapid CSNX during the protocol without the need for any further deep dissection, which could have affected cardiovascular-respiratory measurements.

The first series of experiments assessed the effect of acute CSNX on cardiovascular respiratory parameters during normoxia and acute severe hypoxia, in N ($n = 6$) and CH ($n = 6$) animals. Normoxia was defined as breathing air for N and 12% $F_iO_2$ for CH, delivered by a special premixed gas cylinder (balanced $N_2$). Hypoxic responses were made in 8% $F_iO_2$ for 2 min delivered via a special premixed cylinder (balanced $N_2$) before returning to normoxia. Two initial hypoxic responses were performed to confirm reproducibility and stability. The carotid sinus nerve was subsequently sectioned (CSNX), and following a 10 min period of stabilisation a further 2 min hypoxic exposure was performed. After CSNX if animals displayed prolonged apnoeas during the hypoxic exposure they were returned to normoxia before the full 2 min had elapsed. After experiments animals were killed by an overdose of alfaxalone anaesthesia, death confirmed by cervical dislocation.

The second series of experiments (performed in different rats) investigated the dose-dependent effect of CD73 inhibition on cardiovascular-respiratory function during normoxia and hypoxia in N ($n = 8$) and CH ($n = 7$) animals. Initial baseline measurements were established followed by two control responses to hypoxia (8% $F_iO_2$, 2 min). Once parameters had restabilised in normoxia 160 μg kg$^{-1}$ AOPCP in saline was injected i.v. Normoxic cardiovascular-respiratory measurements were taken 5 min after AOPCP injection, followed by a response to hypoxia (8% $F_iO_2$, 2 min). After recovery back into normoxia this was repeated with 320 and 1120 μg kg$^{-1}$ bolus injections of AOPCP. In a subset of these rats N ($n = 4$) and CH ($n = 6$) animals, to determine any carotid body-dependent effects of CD73 inhibition, an additional dose of AOPCP was given in normoxia (1120 μg kg$^{-1}$) after CSNX. Throughout the experiment the bolus volume dose was consistent at 1 ml kg$^{-1}$ for each concentration of AOPCP. After experiments animals were killed by an overdose of alfaxalone anaesthesia, death confirmed by cervical dislocation.

In both series of experiments data were plotted as 10 s average, every 10 s for time courses. Due to the biphasic hypoxic response, which rapidly peaks and then either plateaus or begins to decline, hypoxic comparisons were made at two different time periods. These included a 10 s average during the rising phase (30–40 s after hypoxic induction) and following stabilisation/roll off (110–120 s after hypoxic induction). All physiological data were acquired and processed using a PowerLab and LabChart 8 (ADInstruments).

### Cardiac and haematological measurements

Rat hearts were harvested following confirmation of death after carotid bifurcation removal. Hearts were further dissected to isolate the left and right ventricles (LV and

**Table 1. Haematological and cardiac characteristics of normoxic and chronically hypoxic rats**

|  | N | CH | *P*-value |
|---|---|---|---|
| Animal mass at terminal exp (g) | 246 ± 27 (42) | 223 ± 31 (40) | **0.0007** |
| Growth rate (g day$^{-1}$) | 4.6 ± 1.3 (11) | 2.3 ± 1.1 (39) | **<0.0001** |
| Haematocrit (%) | 46 ± 2 (4) | 55 ± 2.3 (13) | **<0.0001** |
| Haemoglobin (g/dL) | 15.8 ± 0.7 (4) | 19.0 ± 0.7 (13) | **<0.0001** |
| Ventricular septum (mg) | 152 ± 30 (21) | 141 ± 24 (16) | 0.227 |
| Left ventricle (mg) | 367 ± 51 (21) | 325 ± 57 (16) | **0.0233** |
| Left ventricle/body mass | 1.50 ± 0.17 (21) | 1.42 ± 0.29 (16) | 0.315 |
| Right ventricle (mg) | 116 ± 22 (21) | 164 ± 43 (16) | **<0.0001** |
| Right ventricle/body weight | 0.47 ± 0.07 (21) | 0.71 ± 0.15 (16) | **<0.0001** |
| Fulton's index | 0.22 ± 0.03 (21) | 0.36 ± 0.1 (16) | **<0.0001** |

*Note*: Haematological and cardiac adaptations of chronically hypoxic rats (CH, 12% $F_iO_2$ for 9–10 days) compared to normoxic (N) rats reared in air. Haematocrit and haemoglobin levels were measured using an automated blood analyser. The absolute masses of the cardiac ventricles were determined and further normalised relative to body weight and the left ventricle plus septum (Fulton's index). Data are presented as mean ± SD, with the animal numbers shown in brackets. Statistical significance between N and CH groups was assessed using an unpaired Student's *t* test, with *P*-values included in the rightmost column. *P*-values are in bold where $P < 0.05$.

RV) and the septum before being weighed. To identify whether CH had induced typical right ventricular hypertrophy Fulton's index was used: RV mass/(LV + septum mass). To determine haematocrit and haemoglobin levels an arterial blood sample was taken at the end of the *in vivo* experiments, as described above. Blood was drawn into heparinised capillary tubes and placed immediately on ice. Blood samples were analysed using a haematology analyser (Pentra ES 60, Japan), with parameters automatically calculated.

### Data analysis

All data in figures and text are expressed as the mean ± SD. Statistical analysis was performed using an unpaired/paired Student's *t* test, one-/two-way repeated-measures ANOVA with Dunnet's/Bonferroni *post hoc* analysis where appropriate. Where multiple carotid body tissue section or nerve fibre recording measurements were taken from the same experimental animal a nested *t* test or nested one-way ANOVAs was used to take into consideration that some measurements were related and not totally independent. All analyses were performed in GraphPad Prism (version 10), and significance was taken at $P < 0.05$.

## Results

### Ten days exposure to mild CH ($F_iO_2$ of 12%) induces basal carotid body hyperactivity, an augmented hypoxic sensitivity and key alterations in cardiovascular-respiratory parameters

We first sought to characterise a model of CH based on 10 days exposure to an $F_iO_2$ of 12%. This is a relatively mild intensity of hypoxia in line with those seen in COPD patients compared to more commonly used models using more severe levels of hypoxia. CH rats had reduced body mass and exhibited a reduction in growth rate over the 10 days compared to N (Table 1). CH animals had an approximate 20% elevation in haematocrit and a similar 20% rise in haemoglobin concentration. Absolute LV mass was decreased in CH rats, but this difference was not apparent when normalised to body mass. In contrast CH rats exhibited an approximate 40% increase in RV mass and a 60% increase in the RV mass/(LV + septum mass) ratio (Fulton's index), both indicative of considerable RV hypertrophy (Table 1).

In this model CH induced an elevation in TH$^+$ staining in the carotid body, but didn't appear to cause major alterations in carotid body size (Fig. 2*A*–*C*). There was also no evidence of significant vascular expansion, as

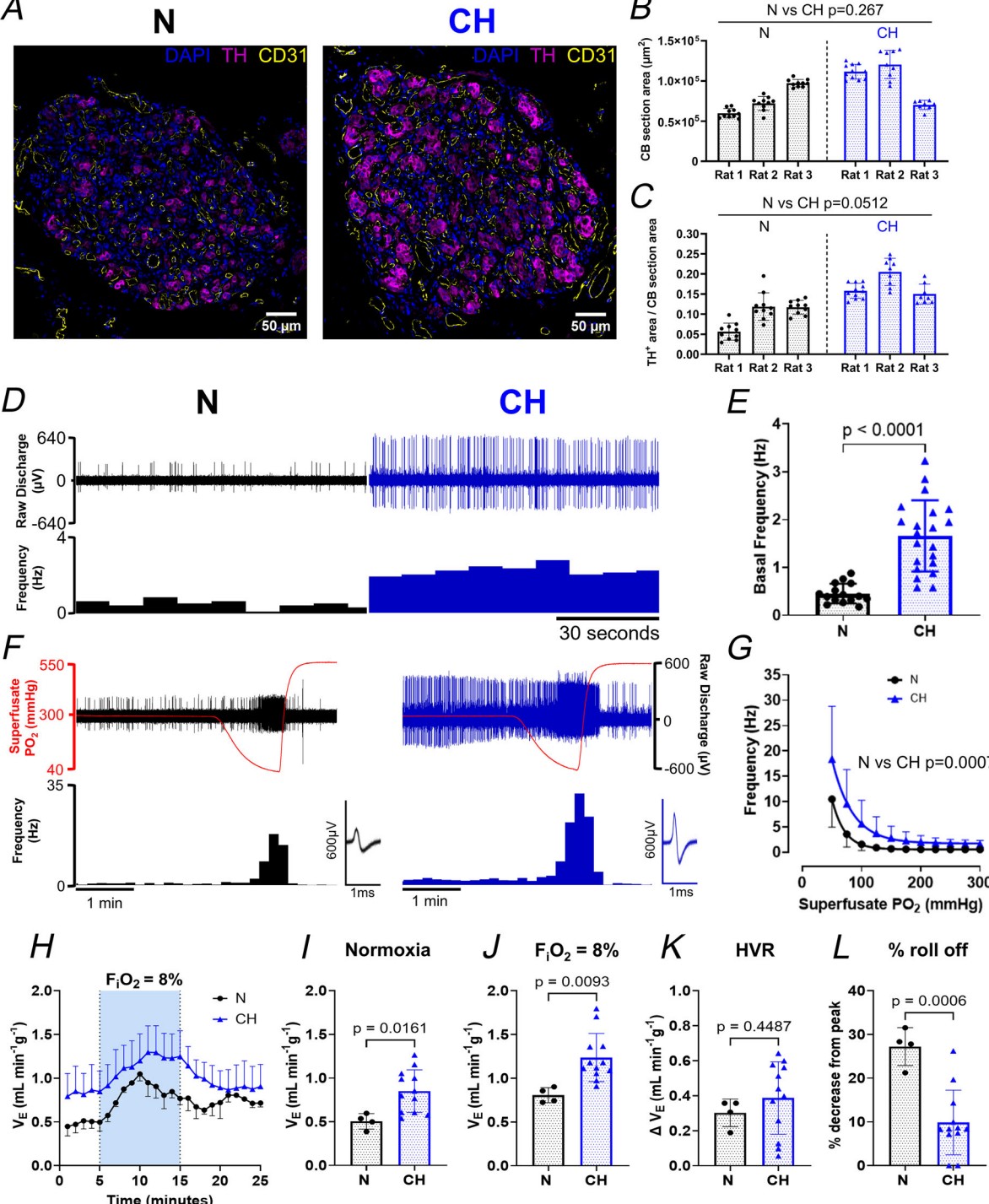

**Figure 2. Mild chronic hypoxia (CH) induces basal carotid body hyperactivity, augmented hypoxic sensitivity and a rise in minute ventilation**

*A*, example confocal immunofluorescence images of carotid body sections from a normoxic (N, left) or chronically hypoxic (CH: $F_iO_2$ 12%, 10 days, right) rat showing tyrosine hydroxylase (TH)-positive type I cells and CD31-positive endothelial cells. TH: 1:500, magenta and CD31: 1:500, yellow. *B*, mean carotid body section area. *C*, mean TH signal density for each animal. Individual data points represent single sections. In *B* and *C*, for N, *n* = 30 sections from three animals, for CH, *n* = 28 sections from three animals. Nested *t* tests. *D*, example traces of basal chemoafferent activity from N (left) and CH (right). Raw discharge (μV) is shown in the upper panels, and frequency histograms

(Hz) are shown in the lower panels. Axes scales are the same for N and CH. *E*, mean basal frequency in N and CH. Individual data points represent a single fibre. Nested *t* test. For N, *n* = 15 fibres from eight rats and for CH, *n* = 21 fibres from nine rats *F*, example recordings of chemoafferent responses to graded hypoxia for N (left) and CH (right). Raw discharge (µV) is shown in the upper panels (axis on the right), and the superfusate $PO_2$ (mmHg) is overlaid in red (axis on the left). Frequency histograms (Hz) are shown in the lower panels (axis on the left). All axes scales are the same for N and CH. Inset: overdrawn action potentials showing single-fibre discrimination. *G*, mean chemoafferent frequency responses to graded hypoxia for N and CH. Two-way repeated-measures ANOVA. N, *n* = 15 fibres from eight rats and for CH, *n* = 21 fibres from nine rats. *H*, ventilatory time course for N and CH showing normoxic ventilation and responses to hypoxia. Normoxic ventilation was measured at $F_iO_2$ of 21% for N and 12% for CH. Hypoxic responses were measured during $F_iO_2$ of 8%. *I–L*, mean normoxic minute ventilation (*I*), hypoxic minute ventilation (*J*), hypoxic ventilatory response (HVR) (*K*) and percentage ventilatory roll off from peak hypoxia (*L*) for N and CH. For *I–L* single data points represent individual rats. Unpaired Student's *t* tests. For N, *n* = 4 rats and for CH, *n* = 12 rats. All data are presented as mean ± SD.

quantification of immunoreactivity for the vascular endothelial marker CD31 showed no difference between N and CH ($CD31^+$ area/carotid body area, N: 0.080 ± 0.02 *vs.* CH: 0.081 ± 0.01, nested *t* test, *P* = 0.952). *Ex vivo* recordings of carotid body chemoafferent activity identified a 2–3-fold elevation in basal frequency in CH (Fig. 2*D* and *E*). During graded hypoxia chemoafferent frequency was augmented in CH, and the $PO_2$-response curve for CH was right-shifted (Fig. 2*F* and *G*). This shift was quantified by measuring the superfusate $PO_2$ required to achieve a discharge frequency of 5 Hz, which was higher in the CH group (N: 65 ± 16 mmHg *vs.* CH: 100 ± 39 mmHg, *P* = 0.0112, nested *t* test). Thus as well as having basal hyperactivity, CH carotid bodies displayed an increase in hypoxic sensitivity. *In vivo* measurements of breathing in awake animals showed that CH rats had a higher normoxic minute ventilation, which was sustained throughout hypoxia (Fig. 2*H–J*). Although peak HVR was not different between groups, the percentage roll off was lower in CH, suggesting an increased ability to sustain the ventilatory response.

## CH increases the proportion of CD73+TH+ cells in the carotid body

Figure 3*A* illustrates confocal immunofluorescence images from N and CH rat carotid body sections stained with antibodies targeting CD73 (Fausther et al., 2012) and TH. In both N and CH CD73 strongly co-localised with TH, suggesting that CD73 protein is expressed in almost every mature type I cell ($CD73^+TH^+$) (Fig. 3*A*). However in N the proportion of $CD73^+$ cells in the carotid body exceeded that of $TH^+$ cells by approximately 10%–15%, identifying a second $CD73^+$ cell population that did not express TH protein at measurable levels ($CD73^+TH^-$) (Fig. 3*A–C*). A considerable number of these $CD73^+TH^-$ cells were located adjacent to the $CD73^+TH^+$ cells within the same cluster and had a similar round morphology (Fig. 3*A*). Following CH the total percentage of $CD73^+$ positive cells did not change (Fig. 3*A* and *C*). However the proportion of $CD73^+TH^+$ cells increased as evidenced by a higher Manders' coefficient (N: 41 ± 6% *vs.* CH: 69 ± 7%

CD73, overlap with TH, *P* = 0.0497, nested *t* test) (Fig. 3*A* and *D*). This reveals that the new mature TH-positive type I cells in CH carotid bodies also positively express CD73.

## Basal carotid body hyperactivity and heightened hypoxic sensitivity in CH is reversed by inhibitory targeting of CD73

To explore if CD73 is responsible for basal carotid body hyperactivity in CH *ex vivo* nerve recordings for N and CH carotid bodies were made in the presence of AOPCP, a CD73 inhibitor. Over a range of 10 nM to 330 µM AOPCP caused a concentration-dependent decrease in chemoafferent frequency in both N and CH, an effect that was rapidly reversible (Fig. 4*A*). In N higher concentrations of AOPCP reduced basal chemoafferent frequency to less than 0.25 Hz (Fig. 4*B*). Despite the elevated basal discharge in CH higher concentrations of AOPCP still severely depressed chemoafferent frequency to less than 0.25 Hz in six/seven fibres (Fig. 4*C*). As such the absolute decrease in basal frequency caused by AOPCP was greater in CH (Fig. 4*D*). When normalised a greater than 90% reduction in basal activity was apparent for both N and CH (Fig. 4*E*). IC50s for N and CH were 1.4 µM (95% confidence interval (CI), 0.97–1.9 µM) and 7.8 µM (95% CI, 4.9–12.4 µM), respectively, (Fig. 4*E*). Therefore basal chemoafferent hyperactivity in CH can be alleviated by inhibition of CD73.

To test if CH basal carotid body hyperactivity could be due to exaggerated adenosine sensitivity responses to 100 µM adenosine were recorded in N and CH (Fig. 4*F*). Adenosine increased firing frequency in both groups (Fig. 4*G*). CH had a higher frequency at baseline, which persisted in the presence of adenosine (Fig. 4*F* and *G*). However both groups exhibited a similar increase in absolute frequency in response to adenosine (Fig 4*H*). As such the relative percentage increase was significantly blunted in CH (Fig. 4*I*), suggesting that CH carotid bodies do not have an elevated adenosine sensitivity.

Next we investigated whether CD73 contributed to the augmented carotid body hypoxic sensitivity

in CH. Experiments were performed using two fixed concentrations of AOPCP – 15 and 100 μm. Figure 5*A* shows an example recording taken from a CH carotid body, demonstrating that AOPCP caused a concentration-dependent decrease in the response to

graded hypoxia, which was reversible. In CH 15 μm AOPCP decreased, but did not abolish, chemoafferent responses to hypoxia (Fig. 5*B*); 15 μm AOPCP induced a left shift in the CH $PO_2$ response curve as evidenced by a reduction in the superfusate $PO_2$ required to elicit a firing

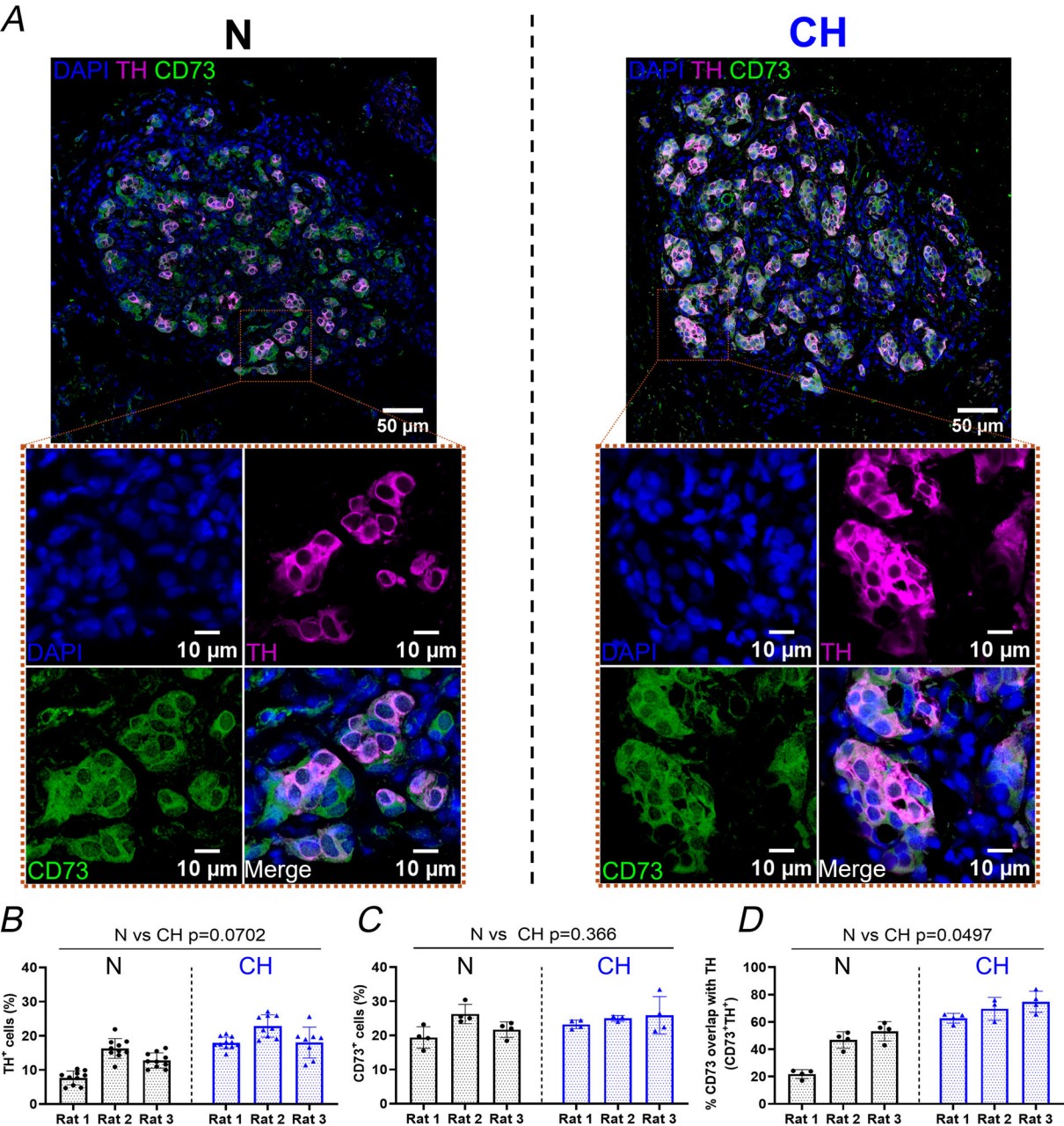

**Figure 3. Chronic hypoxia (CH) increases the CD73⁺TH⁺ cell population within the carotid body**
*A*, example confocal immunofluorescence images from normoxic (N, left) and CH (right) carotid body sections triple stained with CD73 (1:100, green), tyrosine hydroxylase (TH, 1:500, mature type I cell maker, magenta) and DAPI. Insets (dotted boxes): magnified regions of interest. *B*, percentage of TH⁺ cells in N and CH. N, *n* = 30 sections from three animals, CH, *n* = 28 sections from three animals, nested *t* test. *C*, percentage of CD73⁺ cells in N and CH. N, *n* = 12 sections from three animals, CH, *n* = 11 sections from three animals, nested *t* test. *D*, percentage CD73⁺ overlap with TH⁺ in N and CH. Overlap was quantified using Mander's coefficient. N, *n* = 12 sections from three animals, CH, *n* = 11 sections from three animals, nested *t* test. For *B–D* data presented as mean ± SD for each rat, with individual data points representing measurements from a single carotid body section.

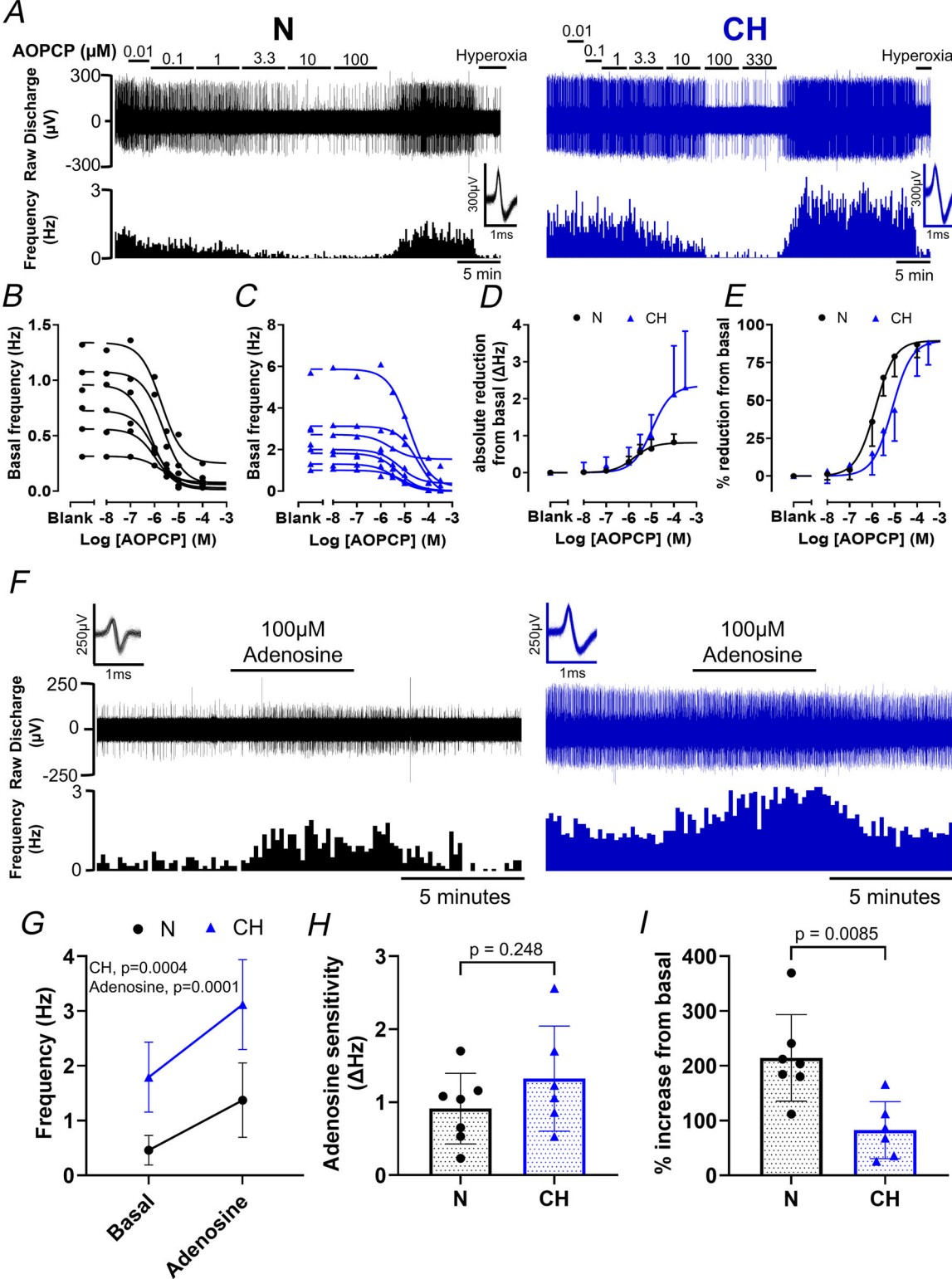

**Figure 4. Basal chemoafferent carotid body hyperactivity in chronic hypoxia (CH) can be fully reversed by pharmacological inhibition of CD73 using AOPCP**

*A*, example recordings of chemoafferent activity from normoxic (N – left) and CH (right) carotid bodies showing the concentration-dependent effects of AOPCP (CD73 inhibitor, 10 nᴍ–330 μᴍ). Raw discharge is shown (upper) along with 10 s frequency histograms (lower). Inset: overdrawn action potentials for single-fibre discrimination. *B* and *C*, each individual AOPCP concentration-response curve replicate for N and CH. *D*, mean AOPCP

concentration-response curves for N and CH expressed as absolute reduction in basal frequency. *E*, mean AOPCP concentration-response curves for N and CH expressed as a percentage reduction from basal frequency. Data presented as mean $\pm$ SD. For N, $n = 6$ fibres from five rats and for CH, $n = 7$ fibres from five rats. *F*, representative raw recordings of chemoafferent activity in response to 100 $\mu$M adenosine for N and CH. *G*, comparison of mean basal frequency and frequency in the presence of 100 $\mu$M adenosine in N and CH. *H*, absolute adenosine sensitivity in Hz (adenosine-basal) for N and CH. *I*, normalised adenosine sensitivity (expressed as percentage increase) for N and CH. Data presented as mean $\pm$ SD with individual data points representing a single fibre. For N, $n = 7$ fibres from six rats and for CH, $n = 6$ fibres from six rats. Nested *t* test.

frequency of 5Hz (Fig. 5*B* and *D*). These effects were magnified in the presence of a higher concentration of AOPCP (100 $\mu$M), such that the hypoxic response curve for a CH carotid body in the presence of 100 $\mu$M was almost indistinguishable from that of an N carotid body (Fig. 5*C* and *E*).

### Transmembrane adenylyl cyclase activity contributes to basal carotid body chemoafferent hyperactivity and exaggerated responses to hypoxia in CH

CD73 signalling is predicted to involve the activation of tmAC, thereby inducing carotid body hyperactivity in CH. To test this we measured chemoafferent activity in the presence of SQ22536 (a tmAC inhibitor, 0.1–500 $\mu$M). In all fibres tested in both N and CH, SQ22536 reduced, but did not abolish, basal discharge with a measurable effect being observed between 1 $\mu$M and 10 $\mu$M and a maximal effect peaking at around 300 $\mu$M (Fig. 6*A*–*C*). The absolute decrease in basal frequency was greater in CH (Fig. 6*D*). When normalised to percentage, SQ22536 produced an approximate 40%–60% decrease in basal discharge that was similar in both N and CH (Fig. 6*E*). IC50s for N and CH were 22.1 $\mu$M (95% CI; 11–41 $\mu$M) and 46.4 $\mu$M (95% CI; 21–97 $\mu$M), respectively. Acute hypoxic sensitivity was subsequently evaluated in the presence of 171 $\mu$M SQ22536 – a concentration producing approximately 90% maximum efficacy. The addition of 171 $\mu$M SQ22536 to a CH carotid body decreased the chemoafferent frequency throughout graded hypoxia (Fig. 6*F*). SQ22536 caused a left shift in the CH hypoxic response curve as evidenced by a decrease in the PO$_2$ required to achieve a frequency of 5 Hz (Fig. 6*F* and *G*). This led to the observation that the hypoxic response curve of a CH carotid body plus SQ22536 almost resembled that of an N carotid body (Fig. 6*F*), that is, SQ22536 attenuated the exaggerated hypoxic sensitivity of a CH carotid body back to pre-adapted levels.

### Complete bilateral CSNX in CH animals decreases arterial blood pressure in normoxia but abolishes the ventilatory response to acute severe hypoxia

To compare the relative importance of the carotid body in cardiovascular respiratory control in N and CH animals measurements of minute ventilation, mABP and HR

were made before and after acute CSNX. Example raw traces and time courses of these parameters measured in normoxia and in response to 2 min of acute hypoxia (F$_I$O$_2$ 8%) before and after CSNX are presented in Fig. 7*A*–*D*. In normoxia CSNX decreased minute ventilation in both groups, and this was exaggerated in CH ($\Delta$V$_E$ N: $-0.161$ $\pm$ 0.03 *vs.* CH: $-0.332 \pm 0.11$ ml min$^{-1}$g$^{-1}$, $P = 0.0039$, unpaired Student's *t* test) (Fig. 7*B*). After CSNX minute ventilation was consistent in N and CH, confirming that increased normoxic ventilation in CH is carotid body dependent (Fig. 7*B*). In normoxia acute CSNX decreased mABP in both groups to a similar extent ($\Delta$ mABP, N: $-24$ $\pm$ 18 *vs.* CH: $-26 \pm 30$ mmHg, $P = 0.9$, unpaired Student's *t* test) (Fig. 7*C*). In contrast acute CSNX had no effect on HR in either N or CH in normoxia.

In N animals the impact of CSNX on cardiovascular respiratory responses to acute severe hypoxia was striking. These animals were unable to evoke a substantial ventilatory response and, in some instances, exhibited prolonged apnoeas, as demonstrated in Fig. 7*A*. N animals also displayed an exaggerated fall in mABP and HR (Fig. 7*A*–*D*). In some instances the hypoxic stimulus was removed (animals returned to breathe normoxia) before the 2 min had elapsed to avoid complete cardiovascular-respiratory collapse. Although all CH animals were able to tolerate the full 2 min of severe hypoxia following CSNX, these animals also failed to mount any significant HVR and displayed a fall in HR (Fig. 7*A*–*D*). Thus although complete bilateral CSNX does effectively decrease ABP in CH (and N) in normoxia, it greatly increases the vulnerability to acute severe hypoxia.

### *In vivo* pharmacological inhibition of CD73 in CH animals causes a dose-dependent decrease in mABP in normoxia without abolishing the ability to respond to severe hypoxia

Next we compared the effect of *in vivo* blockade of CD73 on cardiovascular respiratory parameters in N and CH. Animals were injected i.v. with three sequential doses of AOPCP, 160, 320 and 1120 $\mu$g kg$^{-1}$, with each bolus delivered in the same volume of saline. In normoxia the highest dose of AOPCP produced a modest (5%–10%) fall in minute ventilation, and this was only apparent in CH animals (Table 2). In contrast AOPCP caused a dose-dependent fall in mABP in both N and CH, with

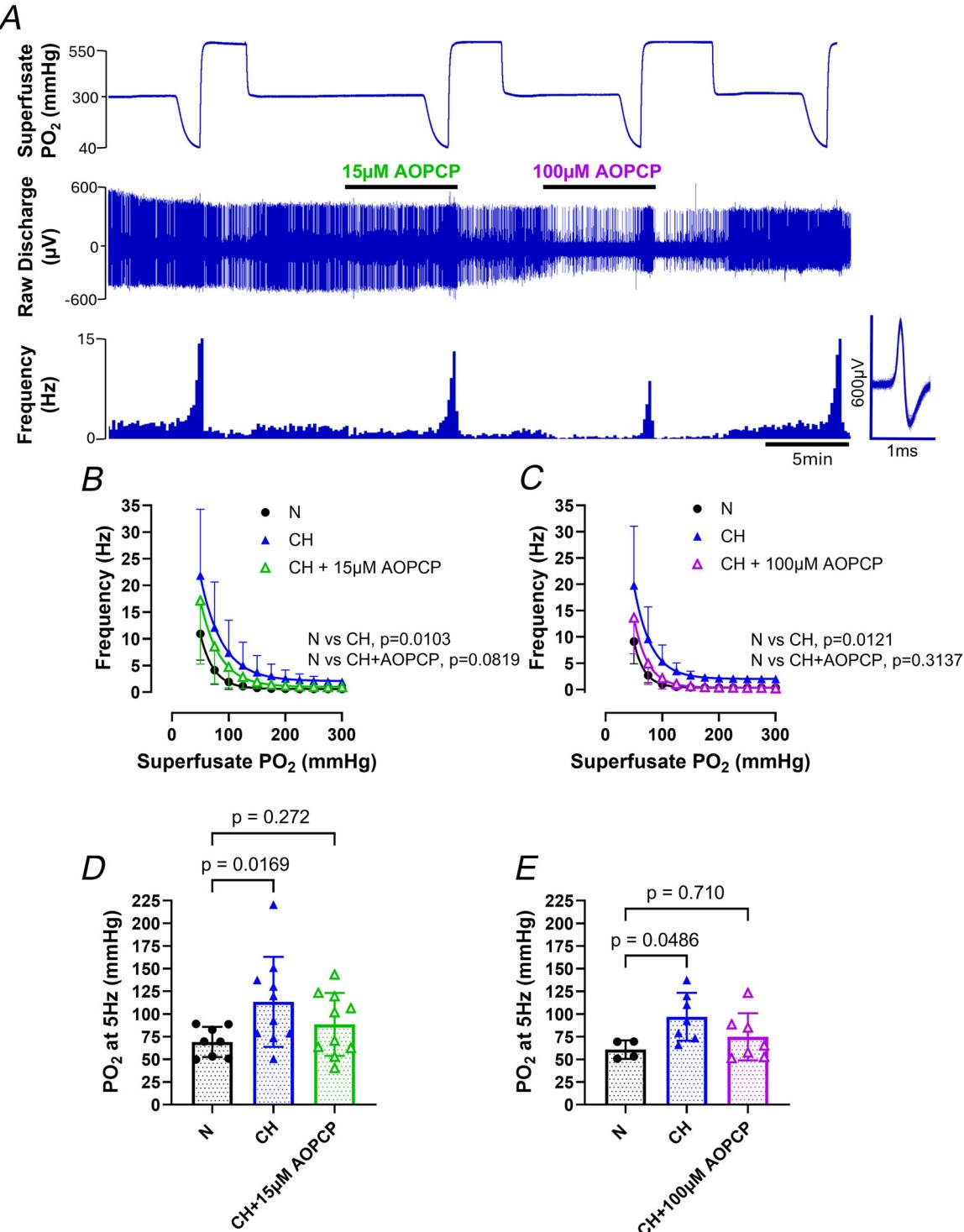

**Figure 5. Pharmacological inhibition of CD73 normalises the exaggerated hypoxic sensitivity of CH carotid bodies**

*A*, raw recording of chemoafferent responses to graded hypoxia from a chronically hypoxic (CH) carotid body showing the effect of 15 and 100 μM AOPCP (CD73 inhibitor). Raw discharge (upper) and frequency histograms (lower). Inset: overdrawn action potentials from a single fibre. *B* and *C*, mean $PO_2$ response curves for N, CH and CH+AOPCP at 15 μM (*B*) and 100 μM (*C*). Two-way repeated measures ANOVA. *D* and *E*, mean $PO_2$ required to elicit a discharge frequency of 5 Hz for N, CH and CH+AOPCP at 15 μM (*D*) and 100 μM (*E*). Nested one-way ANOVA with Dunnett's *post hoc* analysis. All data expressed as mean ± SD, with individual data points representing

a single fibre. For *B* and *D*, N, *n* = 8 fibres from eight rats; CH, *n* = 10 fibres from eight rats; and CH+15 μM AOPCP, *n* = 10 fibres from eight rats. For *C* and *E*, N, *n* = 4 fibres from four rats; CH, *n* = 7 fibres from five rats; and CH+100 μM AOPCP, *n* = 7 fibres from five rats.

**Table 2. The effect of pharmacological inhibition of CD73 using incremental doses of AOPCP on cardiorespiratory measurements in normoxic and chronically hypoxic rats**

| | N | | | | |
|---|---|---|---|---|---|
| | Control | AOPCP (μg kg$^{-1}$) | *P*-value (AOPCP *vs.* control) | Difference (AOPCP-control) | *P*-value (difference *vs.* 160 μg kg$^{-1}$) |
| Minute ventilation (ml min$^{-1}$ g$^{-1}$) | 0.530 ± 0.05 | **160** | 0.540 ± 0.08 | 0.561 | 0.010 ± 0.05 | - |
| | 0.531 ± 0.04 | **320** | 0.549 ± 0.06 | 0.0716 | 0.018 ± 0.03 | 0.792 |
| | 0.525 ± 0.05 | **1120** | 0.544 ± 0.05 | 0.236 | 0.019 ± 0.03 | 0.961 |
| mABP (mmHg) | 128 ± 16 | **160** | 113 ± 11 | **0.0017** | −15 ± 9 | - |
| | 130 ± 11 | **320** | 107 ± 13 | **0.0012** | −23 ± 12 | 0.0692 |
| | 128 ± 11 | **1120** | 95 ± 14 | **0.0005** | −33 ± 15 | **0.0041** |
| Heart rate (beats min$^{-1}$) | 444 ± 32 | **160** | 444 ± 32 | 0.922 | 0 ± 11 | - |
| | 435 ± 26 | **320** | 430 ± 28 | 0.413 | −5 ± 17 | 0.432 |
| | 429 ± 21 | **1120** | 402 ± 31 | **0.0045** | −27 ± 19 | **0.0118** |
| | CH | | | | |
| | **Control** | **AOPCP (μg kg$^{-1}$)** | **P-value (AOPCP *vs.* control)** | **Difference (AOPCP-control)** | **P-value (difference *vs.* 160 μg kg$^{-1}$)** |
| Minute ventilation (ml min$^{-1}$ g$^{-1}$) | 0.736 ± 0.07 | **160** | 0.734 ± 0.06 | 0.932 | −0.002 ± 0.05 | - |
| | 0.748 ± 0.08 | **320** | 0.742 ± 0.08 | 0.333 | −0.006 ± 0.02 | 0.947 |
| | 0.765 ± 0.07 | **1120** | 0.717 ± 0.06 | **0.0037** | −0.048 ± 0.03 | 0.138 |
| mABP (mmHg) | 85 ± 18 | **160** | 81 ± 13 | 0.63 | −4 ± 18 | - |
| | 104 ± 6 | **320** | 79 ± 10 | **0.0056** | −25 ± 15 | **<0.0001** |
| | 108 ± 7 | **1120** | 69 ± 12 | **0.0003** | −39 ± 14 | **0.0022** |
| Heart rate (beats min$^{-1}$) | 444 ± 35 | **160** | 437 ± 32 | 0.226 | −7 ± 13 | - |
| | 452 ± 32 | **320** | 432 ± 31 | **0.0120** | −19 ± 14 | **0.0214** |
| | 454 ± 24 | **1120** | 402 ± 39 | **0.0038** | −52 ± 30 | **0.0068** |

*Note*: AOPCP was given as a bolus injection at three incremental doses – 160, 320 and 1120 μg kg$^{-1}$ to normoxic (N) and chronically hypoxic (CH) rats. Minute ventilation, mean arterial blood pressure (mABP) and heart rate were measured. Data presented as mean ± SD. For N, *n* = 8 rats, and for CH, *n* = 7 rats. Control *vs.* AOPCP, paired *t* test. To compare differences (AOPCP-control) evoked by the two higher doses (320 and 1120 μg kg$^{-1}$) *vs.* 160 μg kg$^{-1}$ one-way repeated-measures ANOVA with Dunnett's *post hoc* analysis was performed. *P*-values are in bold where *P* < 0.05.

the highest dose (1120 μg kg$^{-1}$) reducing the mABP by approximately 25%–40% (Δ mABP, N: −33 ± 15 *vs.* CH: −39 ± 14 mmHg, *P* = 0.4, unpaired Student's *t* test) (Table 2). In normoxia AOPCP also caused a small dose-dependent decrease in HR in both N and CH, with a fall of approximately 5%–10% at the highest dose (1120 μg kg$^{-1}$) (Δ HR, N: −27 ± 18 *vs.* CH: −52 ± 30 bpm, *P* = 0.07, unpaired Student's *t* test) (Table 2).

To examine whether any of the effects of AOPCP in normoxia were acting via the carotid body a subset of animals underwent CSNX. After stabilisation animals were given a second bolus of the highest dose of AOPCP (1120 μg kg$^{-1}$) in normoxia. Figure 8 shows the time courses for minute ventilation, mABP and HR in response to 1120 μg kg$^{-1}$ AOPCP before and after CSNX. Consistent with earlier findings (Fig. 7) CSNX produced a decrease in minute ventilation and mABP in both N and CH. AOPCP given after CSNX had no effect on normoxic ventilation in CH, suggesting that the small decrease seen before CSNX was carotid body dependent (Fig. 8*A* and *B*). Furthermore the reduction in mABP during normoxia caused by AOPCP was attenuated following CSNX in both N and CH, indicating that much of the action of AOPCP on mABP is mediated via the carotid body (Fig. 8*A* and *C*). In contrast the fall in HR produced by AOPCP in normoxia was similar before and after CSNX identifying a carotid body-independent effect (Fig. 8*A* and *C*).

Animals were also exposed to severe hypoxia (F$_i$O$_2$ 8%, 2 min) in the presence of the highest dose of AOPCP (1120 μg kg$^{-1}$) (Fig. 8*A*, left). In contrast to CSNX (Figs 7 and 8, right) all animals in N and CH were able to tolerate the full 2 min of severe hypoxia. In both N and

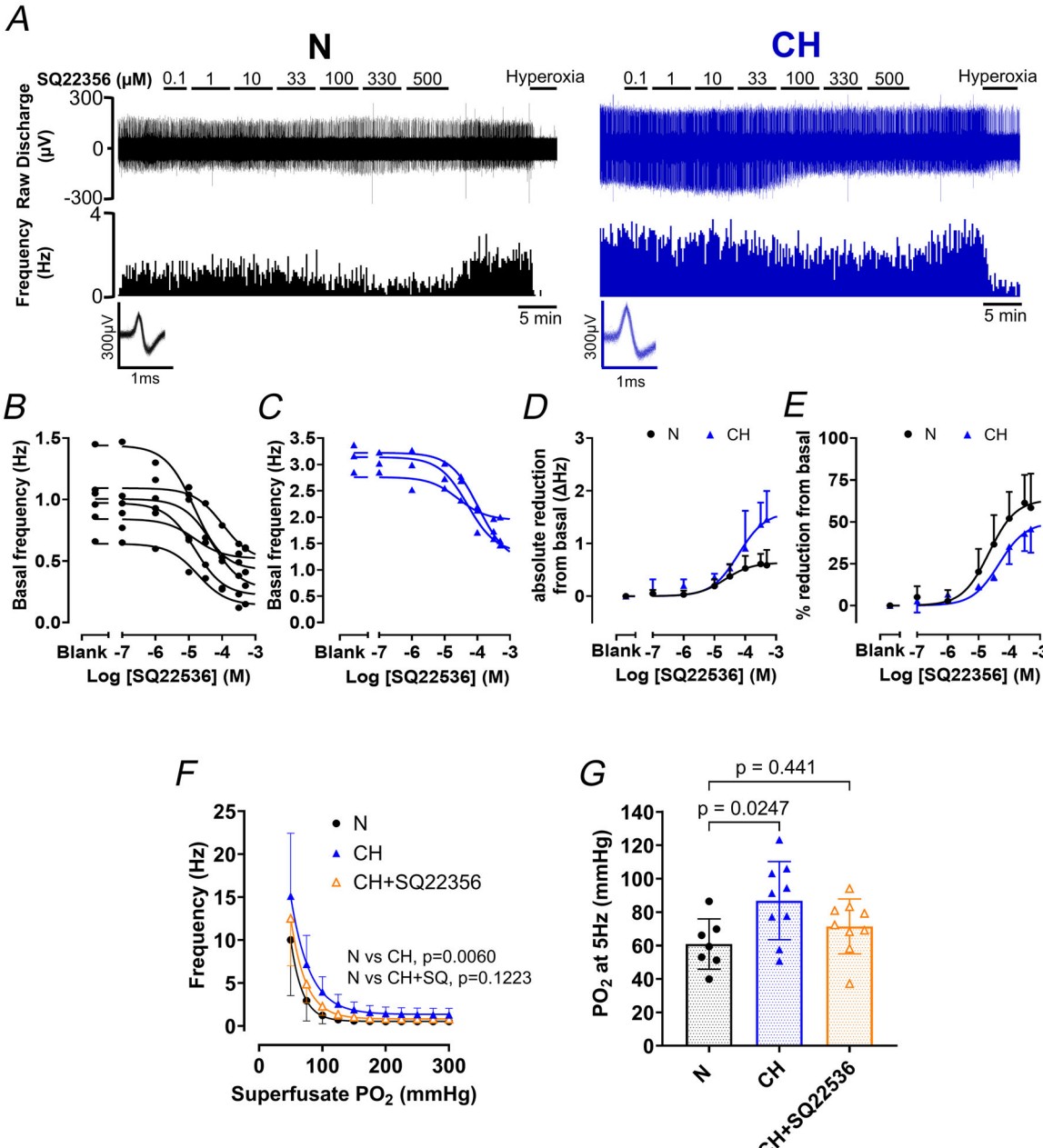

**Figure 6. Transmembrane adenylyl cyclase (tmAC) inhibition normalises chronic hypoxia (CH)-induced carotid body O$_2$ hypersensitivity**

*A*, raw example recordings of basal chemoafferent activity in the presence of SQ22536 (tmAC inhibitor, 0.1 to 500 μM) from a normoxic (N – left) and CH (right) carotid body. Raw discharge (upper) and frequency histograms (lower). Inset overdrawn action potentials for single-fibre discrimination. *B* and *C*, each individual SQ22536 concentration-response curve replicate for N (*B*) and CH (*C*). *D* and *E*, mean SQ22536 concentration-response curves for N and CH expressed as absolute reduction (*D*) and percentage reduction (*E*) in basal frequency. For N, *n* = 6 fibres from six rats, for CH, *n* = 3 fibres from three rats. *F*, mean PO$_2$-response curves for N, CH and CH+SQ22536 (171 μM). Two-way repeated measures ANOVA. *G*, mean PO$_2$ required to elicit a discharge frequency of 5 Hz for N, CH and CH+SQ22536 (171 μM). Nested one-way ANOVA with Dunnett's *post hoc* analysis. For *F* and *G*, N, *n* = 7 fibres from six rats; CH, *n* = 9 fibres from eight rats; CH+SQ22536, *n* = 9 fibres from eight animals. All data presented as mean ± SD. Individual data points represent a single fibre.

CH animals still mounted a substantial HVR in the presence of AOPCP, similar in magnitude to that observed in the absence of the drug (Fig 9*A*). No animals exhibited ventilatory depression or severe apnoea as they did with CSNX (Figs 7–9). Furthermore the fall in blood pressure during severe hypoxia was still apparent in the presence of AOPCP but was not exaggerated (Fig. 9*C*). In N AOPCP did slightly depress the initial elevation in HR during the

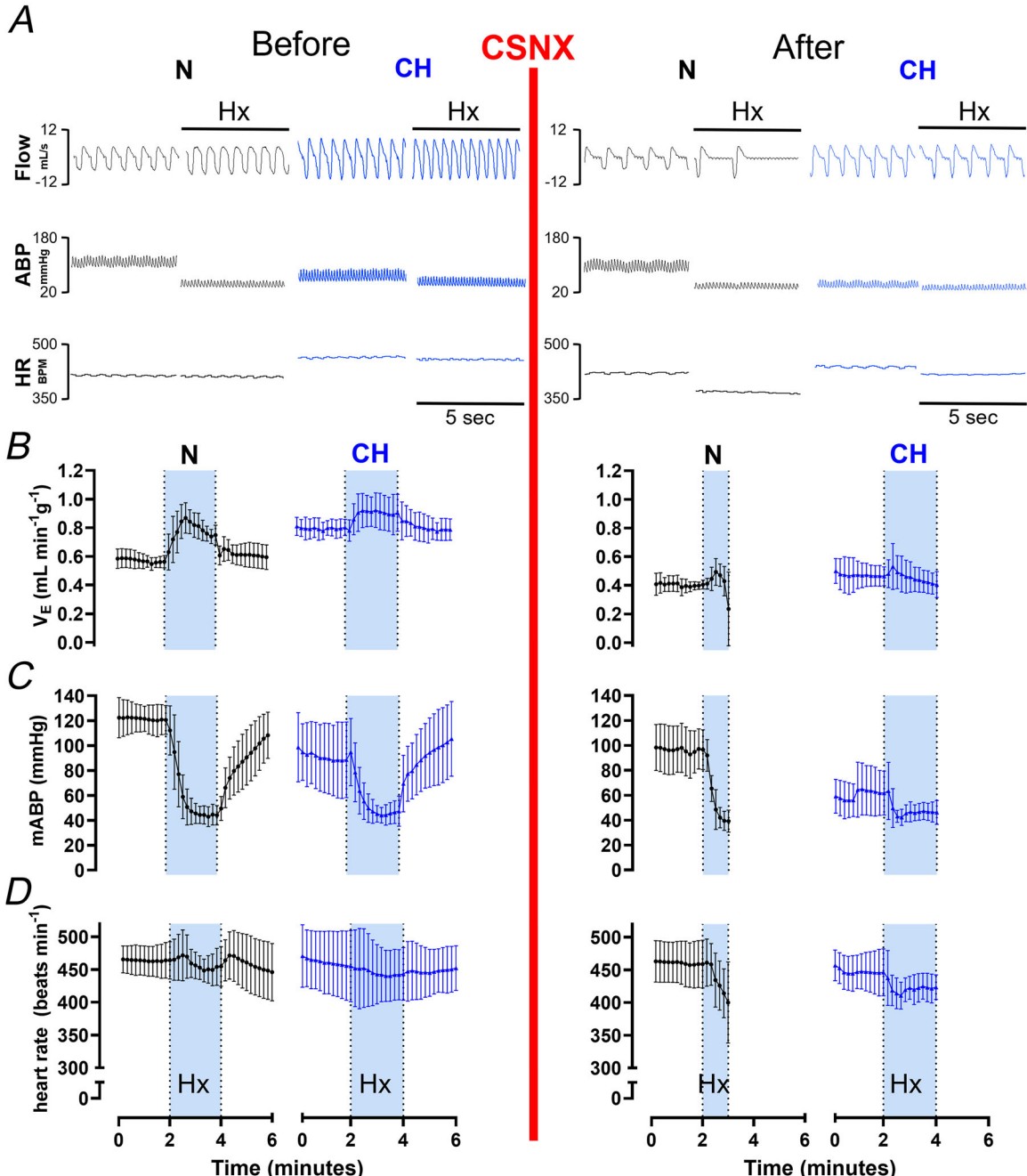

**Figure 7. Acute carotid sinus nerve section (CSNX) in normoxic (N) and chronically hypoxic (CH) animals decreases mean arterial blood pressure (mABP) but increases the risk of cardiovascular-respiratory collapse during exposure to severe hypoxia**

*A*, five second raw traces from an N and a CH rat showing airflow (upper), mABP (middle) and heart rate (HR, lower) measured during normoxia (N-21% $F_iO_2$, CH-12% $F_iO_2$) and severe hypoxia (Hx = $F_iO_2$ 8%; shaded light blue) before and after CSNX (red line). *B–D*, Time courses showing mean minute ventilation (*B*), mABP (*C*) and HR (*D*) during baseline normoxia (N-air, CH-12% $F_iO_2$) and in response to 2 min of acute hypoxia ($F_iO_2$ 8%) before and after CSNX (red line). Data presented as mean ± SD. For N, *n* = 6 rats and for CH, *n* = 6 rats.

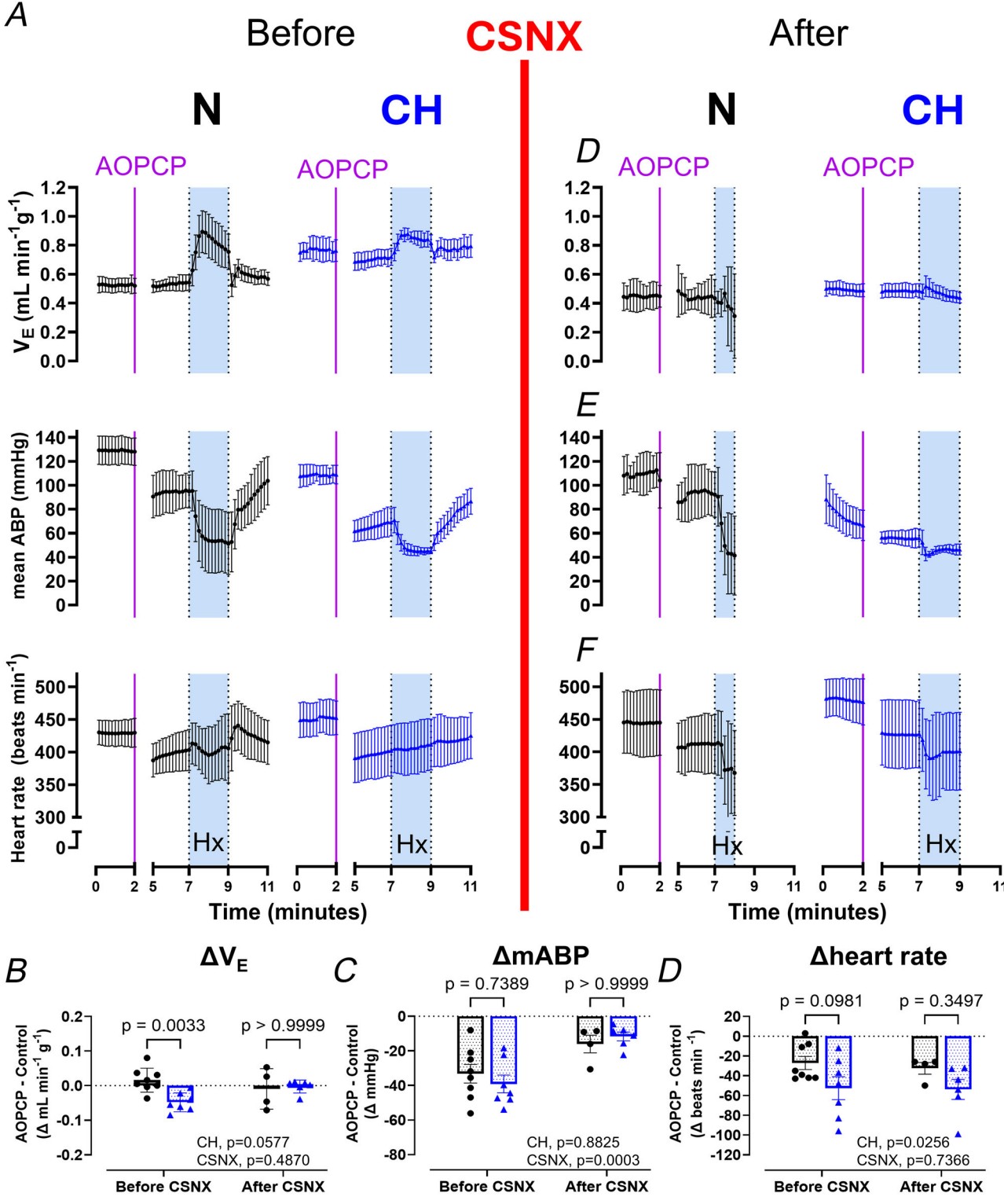

**Figure 8. *In vivo* targeting of CD73 with AOPCP decreases mean arterial blood pressure in normoxia via the carotid body**

*A*, time course showing effect of 1120 µg kg$^{-1}$ AOPCP I.V. on minute ventilation (upper), mean arterial blood pressure (middle) and heart rate (HR, lower) before and after carotid sinus nerve section (CSNX, red line) in normoxic (N) and chronically hypoxic (CH) animals. Time courses include periods of normoxia (N-$F_iO_2$ 21%, CH-$F_iO_2$ 12%) and hypoxia ($F_iO_2$ 8%, shaded light blue). *B–D*, mean changes in minute ventilation (*B*), mABP (*C*) and HR (*D*) caused by AOPCP in normoxia before and after CSNX. For N, *n* = 8 animals before CSNX, *n* = 4 animals after

CSNX. For analysis all data presented as mean ± SD. CH, *n* = 7 animals before CSNX and *n* = 6 animals after CSNX. Two-way ANOVA with Bonferroni *post hoc* analysis.

rising phase of hypoxia, but this was not observed in CH (Fig. 9*D*). With AOPCP in both N and CH the HR did not drop considerably during hypoxia as it did with CSNX (Figs 7, 8 and 9*C*). Thus even at the highest dose of AOPCP the reflex response to severe hypoxia is largely maintained, showing only small signs of attenuation.

## Discussion

### Main findings

The Wistar rat CH model, involving 10 days of exposure to 12% $F_iO_2$, induced basal carotid body chemoafferent hyperactivity, augmented carotid body hypoxic sensitivity and increased minute ventilation. This was associated with an expansion of the $CD73^+TH^+$ cell population in the CH carotid body. *Ex vivo* pharmacological inhibition of CD73 with AOPCP fully reversed basal CH carotid body chemoafferent hyperactivity and normalised the heightened hypoxic sensitivity. These findings were partially replicated with SQ22536, a tmAC inhibitor, suggesting a role for cAMP in CH-induced basal chemoafferent hyperactivity and augmented hypoxic sensitivity. *In vivo* AOPCP lowered normoxic ABP in both N and CH animals via a carotid body-dependent mechanism. Importantly cardiovascular and respiratory reflex responses to severe hypoxia remained largely intact in the presence of AOPCP in CH animals, contrasting greatly with bilateral CSNX. This highlights a new role for CD73 and tmAC in promoting carotid body hyperactivity in response to CH. CD73 inhibition may be a promising therapeutic strategy to reduce carotid body hyperactivity and decrease ABP in CH-related diseases like COPD, without compromising the critical ability to respond to severe hypoxia.

### CH increases the $CD73^+TH^+$ cell population in the carotid body

CD73 is a key enzyme involved in purine metabolism, specifically converting AMP into adenosine. It has been reported that higher levels of CD73 mRNA are expressed in the carotid body compared to other tissues, including brain, super cervical ganglion and petrosal ganglion (Salman et al., 2017). CD73 protein is also expressed in the carotid body, specifically in type I and type II cells (Sacramento et al., 2019; Salman et al., 2017). CD73 functions to control extracellular adenosine concentration following the release of ATP from type I and type II cells (Conde & Monteiro, 2004; Leonard et al., 2018; Murali &

Nurse, 2016). In the normoxic unadapted carotid body our preceding studies revealed that CD73 is important in establishing sensitivity to hypoxia and hypercapnia (Holmes et al., 2015, 2018).

It has been previously shown that whole carotid body CD73 mRNA expression is elevated by CH (Salman et al., 2017). The current study used immunohistochemical techniques to explore changes in the spatial arrangement of CD73 protein and/or modifications in the cellular distribution of CD73 in the adult carotid body, induced by CH. Our data indicate that almost all mature TH-positive type I cells co-express CD73 ($CD73^+TH^+$). However in N animals we identify a distinct population of cells that express CD73 but not TH ($CD73^+TH^-$). Most of these $CD73^+TH^-$ cells are in the same glomeruli directly adjacent to the mature $CD73^+TH^+$ type I cells and have a similar morphology. CH led to a significant expansion of the pool of mature $CD73^+TH^+$ type I cells and a decline in the $CD73^+TH^-$. The mechanism underpinning the increase in $TH^+$ type I cells in the carotid body in response to CH remains contested. One hypothesis proposes an adult stem cell niche composed of quiescent type II cells during normoxia. Under CH these cells differentiate into $Nestin^+$ progenitors and eventually mature type I cells (Pardal et al., 2007). Another hypothesis suggests that type I cell expansion arises from the type I cells themselves. Lineage-tracing experiments have shown that newly dividing $TH^+$ cells originate from a pre-existing $TH^+$ population, which is heavily HIF-2$\alpha$ dependent (Fielding et al., 2018). Similarly Sobrino and colleagues proposed a subpopulation of mitotic cells, termed immature neuroblasts, which express low levels of TH and the marker HNK-1, and upon 24–48 h exposure to hypoxia leads to a rapid increase in the mature type I cell population without carotid body hypertrophy (Sobrino et al., 2018). It is tempting to speculate that the $CD73^+TH^-$ cells we have identified are indeed neuroblasts with weak levels of TH protein expression that were below the detection threshold of the staining protocol. After CH upregulation of TH may have made these cells detectable, which now overlap with the already expressed CD73, leading to the rise in $CD73^+TH^+$ cells. Alternatively some of the increase in $CD73^+TH^+$ population could derive from type II cells, which express CD73 protein (Salman et al., 2017). Determination of the exact nature of the $CD73^+TH^-$ cells warrants further study. However our data do indicate that in the CH-adapted carotid body the newly formed TH-positive type I cells co-express CD73. We acknowledge that CD73 and TH protein localisation data were analysed from a small number of animals, and

therefore future validation will be useful. Furthermore quantification of CD73 protein expression in the different subtypes of cells within the carotid body using western blotting could be an important avenue to explore in the future.

### A new role for CD73 and cAMP in promoting carotid body chemoafferent hyperactivity in CH

Many adaptations have been characterised in carotid body type I cells in response to CH. These include alterations in

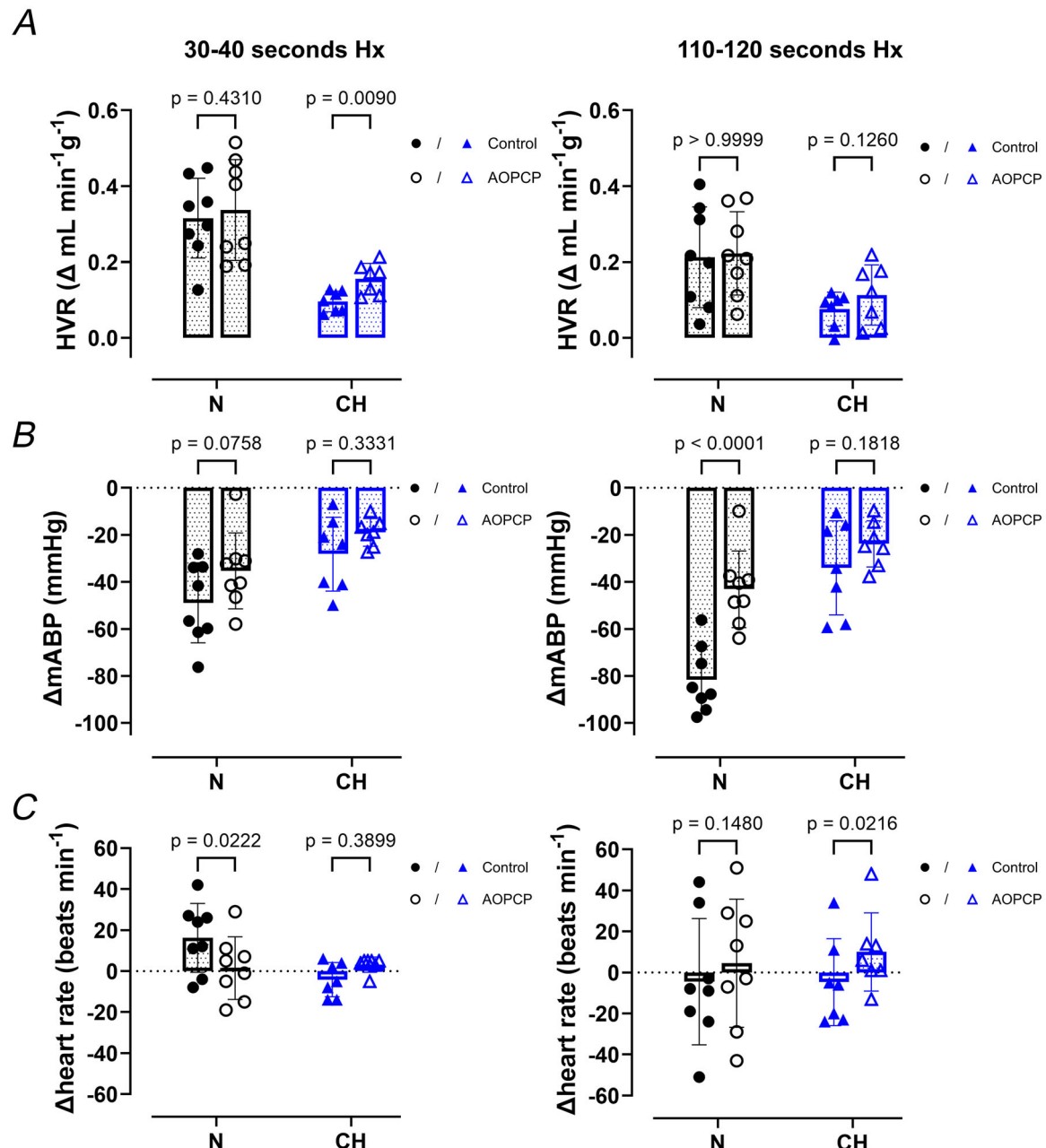

**Figure 9. AOPCP does not abolish cardiovascular-respiratory reflex responses to severe hypoxia in normoxic (N) and chronically hypoxic (CH) rats**

*A–C*, changes in minute ventilation (hypoxic ventilatory response (HVR)) (*A*), mean arterial blood pressure (mABP) (*B*) and heart rate (*C*) in response to severe hypoxia (Hx = $F_iO_2$ 8%) before and after AOPCP administration (1120 μg $kg^{-1}$) for N and CH animals. Changes calculated during the first 30–40 s of hypoxia (rising phase, left panels) and after 110–120 s of hypoxia (steady state, right panels). For N, *n* = 8 rats and for CH, *n* = 7 rats. Data presented as mean ± SD, with each point representing an individual rat. Closed symbols denote control (no drug), and open symbols denote AOPCP. Two-way repeated-measures ANOVA with Bonferroni *post hoc* analysis.

type I cell $K^+$, $Ca^{2+}$ and $Na^+$ currents (Carpenter et al., 1998; Jackson & Nurse, 1997; Ortiz et al., 2009; Spiller et al., 2025; Stea et al., 1995), a more positive resting membrane potential (Spiller et al., 2025) and enhanced catecholamine secretion (Gonzalez-Guerrero et al., 1993; Jackson & Nurse, 1997). The chemoafferent fibres also display plasticity as evidenced by enhanced sensitivity to ATP and ACh (Icekson et al., 2013). There is basal chemo-afferent hyperactivity in CH carotid bodies (Conde et al., 2012b), yet a mechanism underpinning this has remained elusive. Here we provide evidence that exaggerated basal chemoafferent frequency in CH is highly dependent on CD73 and can be fully reversed by AOPCP.

CD73 is an ecto-5'-nucleotidase that converts AMP into adenosine. Upstream of this ATP is initially converted to ADP and then AMP by NTPDases. The importance of CD73 in driving the basal hyperactivity in CH could be due to an enhanced formation of adenosine. Under normal conditions detectable amounts of ATP and adenosine can be recovered in normoxia from whole carotid body tissue (Conde et al., 2012a). In CH type I cells increase the size and density of neurotransmitter-containing dense core vesicles (Fielding et al., 2018; Hodson et al., 2016). The ultrastructure of these dense core vesicles is lighter and eccentrically distributed, suggestive of augmented vesicular neuro-secretion (Fielding et al., 2018). It is therefore plausible that tonic vesicular secretion of ATP during normoxia is elevated in CH, which, after breakdown via NTPDases, provides a greater AMP substrate for CD73. Type II cells may also play a role, as they release ATP in a paracrine manner during hypoxia (Xu et al., 2003). Furthermore CH upregulates NTPDase-3 and CD73 mRNA expression in the rat carotid body while downregulating the equilibrative nucleoside transporter (ENT)-1 and ENT-2 (Salman & Nurse, 2018; Salman et al., 2017). Collectively these point towards a greater pool of extracellular adenosine in the CH carotid bodies mediated via CD73.

Alternatively basal chemoafferent hyperactivity may be dependent on a greater sensitivity to adenosine formed from CD73. Our data demonstrate that the chemo-afferent response to adenosine *per se* was unaffected by CH. In CH-cultured type I cells larger increases in dopamine secretion in response to adenosine are suggested to be due to heightened $A_{2B}$ receptor signalling (Livermore & Nurse, 2013). However a depression in $A_{2A}$ receptor-mediated signalling has been observed in pheochromocytoma-derived, PC12 cells (which are similar to type I cells) following CH (Kobayashi et al., 1998). This reciprocal relationship may explain the lack of increased chemoafferent sensitivity to adenosine following CH seen here. That said $A_{2A}$ receptor knock-out mice do not exhibit baseline ventilatory acclimatisation to short-term sustained hypoxia (Souza & Machado, 2023).

Measurements of carotid body hypoxic sensitivity in these mice would be very interesting.

Although basal carotid body activity in CH was attenuated by AOPCP, chemoafferent frequency was still able to increase during hypoxia, albeit the response being left shifted. Indeed if the hypoxic stimulus is severe enough, the nerve frequency can still rise to a maximal level. This is consistent with our *in vivo* observations that cardiovascular-respiratory reflexes were largely pre-served in the presence of AOPCP when exposed to a single level of severe hypoxia ($F_iO_2$ 8%). As well as providing important safety information these data suggest that in the CH carotid body CD73 and adenosine are both more involved in setting basal chemoafferent activity and the mild-moderate phases of the hypoxic response. The importance of adenosine in regulating the early-phase response to hypoxia in N animals has been reported pre-viously (Conde et al., 2012a).

In the carotid body CD73 is linked to tmAC via adenosine receptors (Conde et al., 2008; Holmes et al., 2015; Monteiro et al., 1996). We show that the tmAC inhibitor, SQ22536, also decreased basal chemoafferent activity in CH and normalised its exaggerated hypoxic sensitivity. Thus tmAC and cAMP appear to contribute to basal chemoafferent hyperactivity and elevated $O_2$ sensitivity in CH. Evidence has directly implicated cAMP in altering the $Na^+$ channel density and long-duration $Ca^{2+}$ components in type I cells following CH (Stea et al., 1995). Plausible downstream cAMP signalling would involve PKA and EPAC, both of which are important in healthy carotid bodies (Rocher et al., 2009; Roy et al., 2013; Thompson & Wyatt, 2011; Xu et al., 2006), but their role in determining carotid body plasticity in response to CH is less clear. Recently it has been reported that cAMP can elevate carotid body hypoxic sensitivity via the activation of CNG ion channels (Peng et al., 2023). A rise in cAMP leading to more persistent stimulation of these CNG ion channels is proposed to be causative of carotid body hyperactivity in response to chronic intermittent hypoxia (Peng et al., 2025). Whether CD73 signalling converges on CNG ion channels in the CH carotid body is an intriguing possibility. That said, the impact of SQ22536 was far less striking than AOPCP. Given that six transmembrane adenylyl cyclase isoforms are expressed in the carotid body it is reasonable to suggest that pharmacological inhibition with SQ25536 may not have completely inhibited all tmAC isoforms (Nunes et al., 2013). Alternatively some of the effects of CD73 inhibition may be independent of a decrease in cAMP and may be related to alterations in extracellular ATP, ADP or AMP.

The greatest impact of both CD73/tmAC inhibition occurred in normoxia and during mild hypoxia suggesting that CD73 and cAMP are important in setting the $PO_2$ threshold needed to initiate a hypoxic response in CH carotid bodies rather than completely preventing it. This

mechanism can be targeted pharmacologically in CH carotid bodies to reduce hypoxic sensitivity back to "normal" non-adapted levels.

## Pharmacological modulation of the carotid body: a safer alternative to CSNX or ablation?

In animal models of carotid body hyperactivity CSNX has been shown to reduce excessive sympathetic outflow, lower blood pressure, and in hypercaloric type 2 diabetic rats restore insulin sensitivity (Abdala et al., 2012; Ribeiro et al., 2013; Sacramento et al., 2017). In chronic intermittent hypoxia models of obstructive sleep apnoea carotid body ablation and CSNX reduce hypertension and autonomic changes (Del Rio et al., 2016; Pereyra et al., 2025). Preliminary human studies suggest similar benefits in patients with heart failure and neurogenic hypertension (Narkiewicz et al., 2016; Niewinski et al., 2017). However surgical intervention has limited long-term efficacy, and there are major safety concerns regarding severe $O_2$ desaturations during sleep and chronic loss of the chemoreflex (Niewinski et al., 2021).

Our model of CH involved exposing animals to an $F_iO_2$ of 12% for 9–10 days. This is a relatively mild stimulus intensity compared to the more commonly used $F_iO_2$ of 10%. However an $F_iO_2$ of 12% was selected to induce milder levels of hypoxia, more in line with those observed in COPD patients. The 9–10 day duration was based on previous studies showing that the increase in carotid body chemoafferent discharge frequency peaks at approximately 9 days of CH and does not rise further (Chen et al., 2002). Importantly our model successfully induced key cardiovascular adaptations seen in COPD patients, including an elevation in haematocrit, haemoglobin and right ventricular hypertrophy. Furthermore it led to a robust rise in basal carotid body chemoafferent activity, an augmentation in carotid body hypoxic sensitivity and a rise in baseline minute ventilation. Therefore we think that this is a valid model to assess cardiorespiratory and carotid body adaptation to mild levels of chronic sustained hypoxia, possibly more relevant to COPD patients.

No studies have yet quantified the cardiovascular-respiratory changes in response to CSNX in the context of CH. Here CSNX caused a decrease in minute ventilation and blood pressure in N and CH with no effect on HR. Notably the reduction in ventilation was more pronounced in CH rats than in N rats, identifying that elevated baseline ventilation in CH is highly carotid body dependent. CSNX also completely abolished the HVR in N and CH, resulting in a striking finding of impending cardiorespiratory collapse in N rats during severe hypoxia. CH animals, on the contrary, although they did not mount any HVR, were able to tolerate the full hypoxic exposure, likely due to CH adaptations, including

elevated haematocrit and haemoglobin (Table 1). We cannot rule out that central adaptations occurred in these animals, thus compensating for a lack of peripheral chemoreceptor input. The inability of CSNX animals to mount a compensatory ventilatory response under these conditions underscores the inherent danger of ablative interventions, especially in those at risk of severe low $O_2$ availability, such as in COPD.

In this context pharmacological modulation of carotid body activity represents a safer and more clinically viable strategy. There have been promising preclinical findings targeting purinergic signalling in the carotid body using $P2X_3$ antagonists in hypertension and heart failure (Lataro et al., 2023; Pijacka et al., 2016). In our model of CH we showed that acute *in vivo* pharmacological inhibition of CD73 in normoxia successfully decreased ABP acting predominantly via the carotid body. This was based on the finding that the fall in mABP caused by AOPCP was attenuated following CSNX. This was also apparent in N animals. However in both N and CH the decrease in mABP by AOPCP was not completely abolished following CSNX, suggestive of additional, albeit more minor, mechanisms. We cannot rule out potential direct actions of AOPCP on the vasculature. Because AOPCP can cross the blood–brain barrier, carotid body-independent effects may involve the modulation of adenosine pathways within the medulla, such as the NTS and RVLM, regions known to integrate sympathetic output (Dale et al., 2002; Thomas & Spyer, 1996). AOPCP also only slightly attenuated normoxic ventilation in CH animals. Adenosine has complex effects on respiratory drive, and reduced adenosine signalling via CD73 inhibition may enhance central respiratory output (Herlenius et al., 1997). Considerable preservation of hypoxic reflex responsiveness with AOPCP, unlike CSNX, is a major advantage, supporting translational potential. A key challenge moving forward will be selective and chronic targeting of CD73 in the carotid body.

## Limitations

The *in vivo* cardiovascular-respiratory experiments were performed on anaesthetised rats using alfaxalone as the maintenance intravenous anaesthetic. This anaesthetic was used as it has been previously shown to preserve carotid body-mediated cardiovascular-respiratory reflexes initiated by hypoxia, hypercapnia and hypoglycaemia (Holmes et al., 2018; Thompson et al., 2016). It also allowed us to assess the impact of CD73 inhibition on cardiovascular-respiratory function before and immediately after CSNX, preventing any adaptation and providing valuable mechanistic information. Although the use of anaesthesia is necessary for such invasive surgical procedures, a degree of central depression will have occurred, leading to some attenuation of

sympathetic-mediated vascular responses to hypoxia. In addition the HVRs measured in both groups appear to be slightly smaller than those measured in our awake unanaesthetised experiments using whole-body plethysmography. Thus the effects of CD73 inhibition on cardiovascular and respiratory function in normoxia and hypoxia that we report here may not absolutely reflect what would occur in an awake and freely moving animal. However the basal minute ventilation was comparable between anaesthetised and unanaesthetised rats. Additionally we acknowledge that although our model enabled us to investigate the mechanistic importance of CD73 in CH, it represents only one aspect of COPD. COPD is a multifactorial disease influenced by systemic inflammation, often triggered by prolonged exposure to noxious chemicals from smoking and, more recently, poor air quality. Additionally many COPD patients have comorbidities such as obesity, which further exacerbate metabolic and respiratory dysfunction. The absence of a hypertensive phenotype in CH rats, despite elevated sympathetic tone reported in similar human CH states (Hansen & Sander, 2003; Heindl et al., 2001), suggests the current CH model does not fully replicate the exact cardiovascular profile of COPD. This may also reflect species differences, whereby local dilator influences in the rat predominate despite an overall increase in sympathetic tone (Walsh & Marshall, 2006).

## Conclusion

Taken together our findings indicate that pharmacological inhibition of CD73 within the carotid body reverses maladaptive chemoafferent activity in normoxia without severely compromising hypoxia-driven reflexes. This distinguishes it from surgical denervation strategies like CSNX, which eliminate both pathological and protective outputs of the carotid body. Given that exaggerated carotid body activity is implicated in a range of chronic diseases, including COPD, hypertension and heart failure, pharmacological strategies that selectively dampen this overactivity offer considerable therapeutic promise. Further studies should aim to optimise drug delivery, assess long-term efficacy and evaluate the safety of such approaches in CH models.

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

## Additional information

### Data availability statement

All original data that support the findings of this study are available from the corresponding authors, A.P.H. and D.N., upon request.

### Competing interests

None declared.

### Author contributions

Experimental concepts and design: A.P.H., P.K., A.M.C. and D.N. Acquisition, analysis or interpretation of data for the work: D.N., A.M.C., D.A. and A.P.H. Drafting the work or revising it critically for important intellectual content: D.N., A.M.C., D.A., P.K. and A.P.H. All authors approved the final version of the manuscript; all authors agreed to be accountable for all aspects of the work in ensuring that questions related to the accuracy or integrity of any part of the work are appropriately investigated and resolved; all persons designated as authors qualify for authorship, and all those who qualify for authorship are listed.

### Funding

This work was supported by the British Heart Foundation (BHF) (FS/PhD/20/29093) to D.N., A.P.H., A.M.C. and P.K. D.A. is supported by a Saudi Arabia Ministry of Education PhD Studentship.

### Acknowledgements

The authors are grateful to colleagues in the Biomedical Services Unit (BMSU) and the Microscopy Facility, Technology Hub at the University of Birmingham for their support, help and guidance on animal procedures and imaging experiments, respectively. Graphical abstract created in BioRender, https://BioRender.com/89jd7kl.

### Keywords

blood pressure, cAMP, carotid body, CD73, chronic hypoxia

## Supporting information

Additional supporting information can be found online in the Supporting Information section at the end of the HTML view of the article. Supporting information files available:

**Peer Review History**

