## [Peer Review History · The Journal of Physiology]

Targeting ecto-5'-nucleotidase (CD73) and cAMP alleviates carotid body hyperactivity and reduces blood pressure in chronically hypoxic rats

Demitris Nathanael, Andrew M Coney, Dhaifallah Saud Alotaibi, Prem Kumar, and Andrew P Holmes
DOI: 10.1113/JP289539

Corresponding author(s): Andrew Holmes (a.p.holmes@bham.ac.uk)

Review Timeline:

Submission Date:	19-Jun-2025
Editorial Decision:	28-Jul-2025
Revision Received:	24-Sep-2025
Accepted:	08-Oct-2025

Senior Editor: Harold Schultz

Reviewing Editor: Ken O'Halloran

Transaction Report:

Dear Dr Holmes,

Re: JP-RP-2025-289539 "Targeting ecto-5'-nucleotidase (CD73) and cAMP alleviates carotid body hyperactivity and reduces blood pressure in chronically hypoxic rats" by Demitris Nathanael, Andrew M Coney, Dhaifallah Saud Alotaibi, Prem Kumar, and Andrew P Holmes.

Thank you for submitting your manuscript to The Journal of Physiology. It has been assessed by a Reviewing Editor and by 2 expert referees and we are pleased to tell you that it is acceptable for publication following satisfactory revision.

REVISION CHECKLIST:

We look forward to receiving your revised submission.

Yours sincerely,

Harold Schultz
Senior Editor
The Journal of Physiology

REQUIRED ITEMS

1) - Author photo and profile. First or joint first authors are asked to provide a short biography (no more than 100 words for one author or 150 words in total for joint first authors) and a portrait photograph. These should be uploaded and clearly labelled together in a Word document with the revised version of the manuscript. See Information for Authors for further details.

2) - Your manuscript must include a complete Additional Information section, including competing interests; funding; author contributions and acknowledgements.

3) - Please upload separate high-quality figure files via the submission form.

4) - Your paper contains Supporting Information of a type that we no longer publish, including supplementary tables and figures. Any information essential to an understanding of the paper must be included as part of the main manuscript and figures. The only Supporting Information that we publish are video and audio, 3D structures, program codes and large data files. Your revised paper will be returned to you if it does not adhere to our Supporting Information Guidelines.

5) - Papers must comply with the Statistics Policy: https://jp.msubmit.net/cgi-bin/main.plex?form_type=display_requirements#statistics.

In summary:

- If $n \leq 30$, all data points must be plotted in the figure in a way that reveals their range and distribution. A bar graph with data points overlaid, a box and whisker plot or a violin plot (preferably with data points included) are acceptable formats.
- If $n > 30$, then the entire raw dataset must be made available either as supporting information, or hosted on a not-for-profit repository, e.g. FigShare, with access details provided in the manuscript.
- 'n' clearly defined (e.g. x cells from y slices in z animals) in the Methods. Authors should be mindful of pseudoreplication.
- All relevant 'n' values must be clearly stated in the main text, figures and tables.
- The most appropriate summary statistic (e.g. mean or median and standard deviation) must be used. Standard Error of the Mean (SEM) alone is not permitted.
- Exact p values must be stated. Authors must not use 'greater than' or 'less than'. Exact p values must be stated to three significant figures even when 'no statistical significance' is claimed.

6) - Please include an Abstract Figure file, as well as the Figure Legend text within the main article file. The Abstract Figure is a piece of artwork designed to give readers an immediate understanding of the research and should summarise the main conclusions. If possible, the image should be easily 'readable' from left to right or top to bottom. It should show the physiological relevance of the manuscript so readers can assess the importance and content of its findings. Abstract Figures should not merely recapitulate other figures in the manuscript. Please try to keep the diagram as simple as possible and

without superfluous information that may distract from the main conclusion(s). Abstract Figures must be provided by authors no later than the revised manuscript stage and should be uploaded as a separate file during online submission labelled as File Type 'Abstract Figure'. Please also ensure that you include the figure legend in the main article file. All Abstract Figures should be created using BioRender. Authors should use The Journal's premium BioRender account to export high-resolution images. Details on how to use and access the premium account are included as part of this email.

7) - Please ensure that all figures and tables have a title and legend, and that they have been cited within the main article text.

EDITOR COMMENTS

Reviewing Editor:

Thank you for submitting your manuscript to The Journal of Physiology. The paper provides evidence of a novel mechanism contributing to carotid body plasticity evoked by exposure to chronic sustained hypoxia. The article is well written and very well presented with convincing evidence and limitations of the approaches used openly acknowledged.

Two experts have reviewed the manuscript and both are enthusiastic in their general overview of the study recognising the significance of the findings and potential relevance for the wider community interested in carotid body overactivity in disease states. Both referees have provided critiques, which should be very straightforward to address.

Please note that R1 suggests moving a figure to supplemental files, but The Journal does not support supplementary material so these data can be presented as a Figure in the main manuscript. The significance of the data should be clarified to the referee. You included a supplementary Figure relating to the machine learning approach used for image analysis. This Figure can be incorporated into the main manuscript or you can position it in an appendix at the end of the article, but it will remain within a single document and will be published as part of the main manuscript.

Both referees raise concern with sample sizes for some data sets and this point should be carefully addressed.

Senior Editor:

Thank you for the submission of your research article to the Journal of Physiology for consideration. The article has been reviewed by experts in the field and found to be potentially acceptable for publication pending adequate revision to address all of the concerns raised. Please address all comments from the external referees and reviewing editor, as well as address the list of requirements for publication in the journal, including the statistical requirements, as outlined in this letter, the Instructions to Authors online, and the Journal's policy on rigour and reproducibility, found at the link below.

<https://physoc.onlinelibrary.wiley.com/pb-assets/hub-assets/physoc/documents/TJP-Rigour-and-Reproducibility-Requirements-1724673661727.pdf>

The small sample size in some experiments (e.g., Figures 1 and 2) raises concern about adequate statistical analysis to support conclusions based on these data. It is recommended to increase the sample size to meet an adequate power of the analysis as needed.

1. Include in the Ethical Approval section a statement that the investigators understand the ethical principles under which the journal operates and that their work complies with the animal ethics checklist as outlined by the journal.

2. Sample size must avoid pseudo-replication. (using multiple measurements from the same subject as if they were independent replicates, leading to an inflated sample size and potentially misleading statistical results). Clarify how the statistical sample size was determined for each statistical test. Use mixed-effects models or hierarchical models when you have repeated measurements from the same experimental unit.

3. The small sample size in some experiments (e.g., Figures 1 and 2) raises concern about adequate statistical analysis to support conclusions based on these data. It is recommended to increase the sample size to meet an adequate power (beta) of the analysis as needed.

REFeree COMMENTS

Referee #1:

Nathanael et al. evaluated whether the ecto-5-nucleotidase (CD73) participates in the carotid body hyperactivity in rats exposed to chronic sustained hypoxia (CH) for 10 days. They reported that CH increases the proportion of type I cells containing CD73 in the carotid body. Pharmacological targeting of CD73 and transmembrane adenylyl cyclases reduced basal carotid body chemosensory hyperactivity in CH and normalizes the enhanced responses to acute hypoxia in an in vitro CB preparation. Inhibition of CD73 in CH-rats induced a carotid body related decrease in arterial blood pressure in normoxia, without abolishing cardio-respiratory responses to acute severe hypoxia. Thus, authors concluded that CD73 is important in promoting carotid body hyperactivity in CH animals, suggesting that selective targeting of CD73 and cAMP signalling may be useful to ameliorate the carotid body hyperactivity and improve cardiovascular outcomes in chronically hypoxic patients.

The work is original, and the Ms. is clearly written. The study design and robustness of the experimental data is appropriate. The conclusion is supported by several experiments using different techniques. This interesting work is a contribution to the area of research. However, there are some issues that authors need to address to improve Ms. and make it suitable for publication.

Major:

Fig 2. It seems that CH has a small impact on CD7 cells in the carotid body. Figure 2A and 2B showed that there were no significant changes in TH cells or the percentage of CD73-positive cells in the CB. However, the overlap between CD73-positive cells and TH cells yielded a p-value of 0.0049. This result may reflect the limited sample size (n=3) used in the analysis. Why not measure CD73 using qPCR?

Fig 7. Why were the effects of AOPCP before and after CBNX evaluated in anesthetized rats? (rats were initially anaesthetized with 2-5% isoflurane in O₂, and anesthesia was maintained with Alfaxalone). It is well known that anesthesia impairs sympathetic mediated responses including the vascular response to hypoxia. In freely moving rats, hypoxia increases the arterial blood pressure. This is a serious limitation because an important central sympathetic depression will have occurred. The value of these experiments is unclear. Consider presenting the results in a supplemental figure.

What mechanisms led to reduced arterial pressure by AOPCP in N and CH animals? What is the implication that AOPCP did not abolish cardiovascular-respiratory reflex responses to severe hypoxia in both N and CH rats? Is it possible that there are distinct mechanisms involving CD73 and adenosine effects depending on the severity of hypoxia?

Minor:

- 49 carotid body hyperactivity associated with CH. Therefore, the current project is investigated. Work or study seems more appropriate than project.

- 55 within the carotid body. Pharmacological inhibition of CD73 using AOPCP completely. Please define AOPCP, α , β -methylene ADP.

- 56 carotid body chemoafferent hyperexcitability, augmented hypoxic sensitivity and abolished CH-induced basal chemoafferent hyperactivity. Please, define hyperexcitability and hyperactivity. They are sometimes used interchangeably.

- 58-59 adenylyl cyclase (tmAC) inhibitor SQ22536. CD73 antagonism in vivo reduced arterial blood pressure in a carotid body and dose-dependent manner in normoxia, What does mean "in a carotid body and dose-dependent manner in normoxia?"

- 69-73 Carotid body hyperactivity is recognised as a key factor in the development and progression of cardiovascular pathology in multiple illnesses including obstructive sleep apnoea, chronic heart failure, essential hypertension, diabetes, obesity, chronic inflammation and COPD (Incognito et al.; Schultz et al., 2015; Phillips et al., 2018; Shin et al., 2019; Conde et al., 2023; Felipe et al., 2023; Prabhakar et al., 2023).

- Authors cited most relevant works; however, some important contributions are missed (i.e., Abdala AP. *J Physiol*, 590 (17), 4269-4277. doi: 10.1113/jphysiol.2012.237800, Zera et al. *J Physiol*. 2025 603(9):2487-2516. doi: 10.1113/JP285081. Iturriaga R. *J Physiol*. 2023 601(24):5495-5507. doi: 10.1113/JP284112).

- 254 -257 superfused with the same physiological salt solution mentioned above. Basal chemoafferent activity was measured in a PO₂ of 300mmHg and PCO₂ of 40mmHg. This PO₂ was considered normoxic as this level has been shown to generate basal carotid body discharge of 0.25-1.50Hz (Holmes et al., 2014).

- What is the rationale to superfuse the CB with a saline equilibrated at PO₂ of 300 mmHg? The rat CB in vitro superfused preparation work at lower PO₂ levels in the superfusate. Why do authors consider that high PO₂ as normoxia? Why is the frequency of discharge of 0.25 -1.5 Hz considered normoxia? Is it likely that the measured frequency depends on the window used to select action potential?

- 413- 418 During graded hypoxia chemoafferent frequency was augmented in CH and the PO₂-response curve for CH was right-shifted (Fig. 1F and G). This shift was quantified by measuring the superfusate PO₂-required to achieve a discharge frequency of 5 Hz, which was higher in the CH group (N: 65{plus minus}16 mmHg vs CH: 100{plus minus} 39 mmHg, $p=0.0032$, unpaired Student's t-test). Thus, as well as having basal hyperactivity, CH carotid bodies displayed an increase in hypoxic sensitivity.

- How do you define sensitivity and reactivity?

- Fig 3, It seems that the frequency of chemosensory discharges is higher than the one measured (1 Hz in normoxia and 2 in CH-CBs? Why in Fig 3F, the size of the action potential was different?

- In Fig 3B and C, at what PO₂ the effects of CH and AOPCP on the frequency of discharge was significant? Is this important for determining changes in sensitivity?

- In Fig 1D the frequency is expressed in Hz in N, but in CH is expressed in μ V?

- 816-821 Pharmacological modulation of the carotid body: A safer alternative to CSNX or ablation? In animal models of carotid body hyperactivity, CSNX has been shown to reduce excessive sympathetic outflow, lower blood pressure, and in hypercaloric type 2 diabetic rats, restore insulin sensitivity (Abdala et al., 2012; Ribeiro et al., 2013; Sacramento et al., 2017).

- In chronic intermittent hypoxia models of obstructive sleep apnea, CB ablation and CSNX reduce hypertension and autonomic changes (Pereyra K. *Am J Physiol Lung Cell Mol Physiol*. 2025;328(3): L357-L371; Del Rio R. *Hypertension*. 2016;68(2):436-45).

Referee #2:

Upon activation, the carotid body (CB) has been proven to trigger potent respiratory reflexes as well as being one of the most potent stimulants of cardiovascular sympathetic activity. In this remarkable article of integrative physiology, the authors conduct an exhaustive and elegant study focusing on chronic hypoxia (CH)-induced CB hyperactivity, its systemic consequences, and how they can be reversed pharmacologically. They provide an in-depth, comprehensive physiological perspective of the chemoreception field.

In particular, the authors explore the role of CD73, one of the ectonucleotidases expressed in the CB. They demonstrate that basal chemoafferent CB hyperactivity in CH can be fully reversed by the pharmacological inhibition of CD73 using AOPCP. Furthermore, AOPCP normalises the enhanced hypoxic sensitivity of the CB exposed to CH, which is responsible for an exaggerated hypoxic ventilatory response (HVR). Unlike the effects of previous CB treatment described in the literature, this study reveals that, while pharmacological inhibition of CD73 within the CB does reverse maladaptive chemoafferent activity in normoxia, it does not abolish cardiovascular-respiratory reflex responses to severe hypoxia.

This effect seems to be linked to the activity of adenylyl cyclase (AC), which produces an excess of cAMP - the substrate for CD73 - because the inhibition of AC in parallel experiments using SQ-22536 reproduces the effects of AOPCP. The expression of CD73 by CB cells and its increase in type I cells during chronic hypoxia (CH) are well-known facts, as is the role of cAMP and adenylyl cyclase in the CB response to hypoxia, which can be blocked by SQ 22536. [Salman S, Vollmer C, McClelland GB, Nurse CA. Characterization of ectonucleotidase expression in the rat carotid body: regulation by chronic hypoxia. *Am J Physiol Cell Physiol.* 2017 Sep 1;313(3):C274-C284]. And, in my opinion, the authors do not make it sufficiently clear in the discussion.

However, the most important finding is that the authors unveil a novel function of CD73 in promoting CB chemoafferent hyperactivity in CH. Their results provide evidence that the increased basal chemoafferent frequency observed in CH is highly dependent on CD73 and can be completely reversed by AOPCP. As there is no evident enhanced sensitivity to adenosine, the authors hypothesise that a greater pool of extracellular adenosine exists in CH due to the overexpression of CD73. This reinforces the role of ATP/ADP/ adenosine pool in activating the CB tripartite synapse, which includes afferent activation of the CSN and its plasticity, in CH, as described by Colin Nurse and cols.

The study is highly relevant because the CB is not only an oxygen sensor, but also responds acutely to other stimuli besides oxygen, such as glucose. Feedback from blood-borne hormones that regulate metabolism and even ROS and inflammatory mediators also influence its response. Thus, the CB has been shown to be a major driver of the heightened sympathetic activity associated with several cardiometabolic diseases. Therefore, understanding the mechanisms of activation, cellular transduction and intercellular signalling within the CB is crucial for identifying novel drugs and interventions to reduce CB hyperactivity in disease.

Finally, despite the authors' acknowledgment of limitations in their COPD model, the value of this study remains intact, as preclinical models are instrumental in elucidating the physiology and pathophysiology of CB.

MINOR COMMENTS

There is little to say about the 'Methods' section. The experiments are well designed and remarkably well described. However, some concerns have arisen.

- The authors describe the treatment of rats as chronic sustained hypoxia. Since intermittent hypoxia can also be applied chronically, renaming their experimental approach 'chronic sustained hypoxia' (CSH) clarifies this.

- Why is 10 days of hypoxia treatment used in this CH model?

- Another question concerns the total number of animals used in the study and whether they were used for several non-interfering measures.

- Line 202 says: The relatively high pixel intensity of RBC AF and complexity of the images made regular thresholding and other conventional methods to remove this false-positive signal unsuitable. To address this, a machine learning approach was employed using the Python-based software, ilastik (Berg et al., 2019). For training, immunofluorescence images were manually annotated to label RBC pixels. This process was repeated across approximately four images to train the model. Once trained, the software predicted pixel classifications and batch-processed subsequent images.

Given my lack of knowledge in machine learning, I have a question: Is it enough to use four images to train the model?

- Line 408 says: Figure 1, CH induced an elevation in TH positive staining in the carotid body, but didn't appear to cause major alterations in carotid body size. There was also no evidence of significant vascular expansion since quantification of immunoreactivity for the vascular endothelial marker CD31 showed no difference between N and CH.

As the data are based on a small number of animals (N=3), it is difficult to ascertain whether there have been any changes in size or vascularisation, which are two hallmarks of CB plasticity in chronic hypoxia.

-Table I show normalization of LV and RV mass to body mass. However, to normalize LV and RV mass, it would be more accurate to calculate the left or right ventricle-to-tibia length (LV/TL (mg/mm)) ratio because rat body weight changes after CH.

- Figure 7. Part 'B-D' in the figure does not correspond to the text in the legend.

- There are four references that are incomplete:

- Bishop T & Ratcliffe PJ. (2025). HIF2 α : the interface between oxygen-sensing systems in 909 physiology and pathology. *Physiology*.

- Incognito AV, Barioni NO, Sharkey KA & Wilson RJA. Carotid Body Stress And Inflammatory Mediators Uniformly Excite Autonomic Output. *CircRes* 0.

- Sobrino V, Gonzalez-Rodriguez P, Annese V, Lopez-Barneo J & Pardal R. (2018). Fast neurogenesis from carotid body quiescent neuroblasts accelerates adaptation to hypoxia. *EMBO Rep* 19.

- Peng YJ, Nanduri J, Wang N, Su X, Hildreth M & Prabhakar NR. (2025). Signal Transduction Pathway Mediating Carotid Body Dependent Sympathetic Activation and Hypertension by Chronic Intermittent Hypoxia. *Function (Oxf)* 6.

Finally, including a graphical abstract would enrich the presentation of results.

END OF COMMENTS

JP-RP-2025-289539 REVIEW

Upon activation, the carotid body (CB) has been proven to trigger potent respiratory reflexes as well as being one of the most potent stimulants of cardiovascular sympathetic activity. In this remarkable article of integrative physiology, the authors conduct an exhaustive and elegant study focusing on chronic hypoxia (CH)-induced CB hyperactivity, its systemic consequences, and how they can be reversed pharmacologically. They provide an in-depth, comprehensive physiological perspective of the chemoreception field.

In particular, the authors explore the role of CD73, one of the ectonucleotidases expressed in the CB. They demonstrate that basal chemoafferent CB hyperactivity in CH can be fully reversed by the pharmacological inhibition of CD73 using AOPCP. Furthermore, AOPCP normalises the enhanced hypoxic sensitivity of the CB exposed to CH, which is responsible for an exaggerated hypoxic ventilatory response (HVR). Unlike the effects of previous CB treatment described in the literature, this study reveals that, while pharmacological inhibition of CD73 within the CB does reverse maladaptive chemoafferent activity in normoxia, it does not abolish cardiovascular-respiratory reflex responses to severe hypoxia.

This effect seems to be linked to the activity of adenylyl cyclase (AC), which produces an excess of cAMP — the substrate for CD73 — because the inhibition of AC in parallel experiments using SQ-22536 reproduces the effects of AOPCP. The expression of CD73 by CB cells and its increase in type I cells during chronic hypoxia (CH) are well-known facts, as is the role of cAMP and adenylyl cyclase in the CB response to hypoxia, which can be blocked by SQ 22536. [Salman S, Vollmer C, McClelland GB, Nurse CA. Characterization of ectonucleotidase expression in the rat carotid body: regulation by chronic hypoxia. *Am J Physiol Cell Physiol.* 2017 Sep 1;313(3):C274-C284]. And, in my opinion, the authors do not make it sufficiently clear in the discussion.

However, the most important finding is that the authors unveil a novel function of CD73 in promoting CB chemoafferent hyperactivity in CH. Their results provide evidence that the increased basal chemoafferent frequency observed in CH is highly dependent on CD73 and can be completely reversed by AOPCP. As there is no evident enhanced sensitivity to adenosine, the authors hypothesise that a greater pool of extracellular adenosine exists in CH due to the overexpression of CD73. This reinforces the role of ATP/ADP/ adenosine pool in activating the CB tripartite synapse, which includes afferent activation of the CSN and its plasticity, in CH, as described by Colin Nurse and cols.

The study is highly relevant because the CB is not only an oxygen sensor, but also responds acutely to other stimuli besides oxygen, such as glucose. Feedback from blood-borne hormones that regulate metabolism and even ROS and inflammatory mediators also influence its response. Thus, the CB has been shown to be a major driver of the heightened sympathetic activity associated with several cardiometabolic diseases. Therefore, understanding the mechanisms of activation, cellular transduction and intercellular signalling within the CB is crucial for identifying novel drugs and interventions to reduce CB hyperactivity in disease.

Finally, despite the authors' acknowledgment of limitations in their COPD model, the value of this study remains intact, as preclinical models are instrumental in elucidating the physiology and pathophysiology of CB.

Minor comments

There is little to say about the 'Methods' section. The experiments are well designed and remarkably well described. However, some concerns have arisen.

- The authors describe the treatment of rats as chronic sustained hypoxia. Since intermittent hypoxia can also be applied chronically, renaming their experimental approach 'chronic sustained hypoxia' (CSH) clarifies this.
- Why is 10 days of hypoxia treatment used in this CH model?
- Another question concerns the total number of animals used in the study and whether they were used for several non-interfering measures.
- Line 202 says: *The relatively high pixel intensity of RBC AF and complexity of the images made regular thresholding and other conventional methods to remove this false-positive signal unsuitable. To address this, a machine learning approach was employed using the Python-based software, ilastik (Berg et al., 2019). For training, immunofluorescence images were manually annotated to label RBC pixels. This process was repeated across approximately four images to train the model. Once trained, the software predicted pixel classifications and batch-processed subsequent images.*

Given my lack of knowledge in machine learning, I have a question: Is it enough to use four images to train the model?

- Line 408 says: *Figure 1, CH induced an elevation in TH positive staining in the carotid body, but didn't appear to cause major alterations in carotid body size. There was also no evidence of significant vascular expansion since quantification of immunoreactivity for the vascular endothelial marker CD31 showed no difference between N and CH.*

As the data are based on a small number of animals (N=3), it is difficult to ascertain whether there have been any changes in size or vascularisation, which are two hallmarks of CB plasticity in chronic hypoxia.

-Table I show normalization of LV and RV mass to body mass. However, to normalize LV and RV mass, it would be more accurate to calculate the left or right ventricle-to-tibia length (LV/TL (mg/mm)) ratio because rat body weight changes after CH.

- Figure 7. Part 'B-D' in the figure does not correspond to the text in the legend.

- There are **four references** that are incomplete:

- Bishop T & Ratcliffe PJ. (2025). HIF2 α : the interface between oxygen-sensing systems in 909 physiology and pathology. *Physiology*.
- Incognito AV, Barioni NO, Sharkey KA & Wilson RJA. Carotid Body Stress And Inflammatory Mediators Uniformly Excite Autonomic Output. *CircRes* 0.
- Sobrino V, Gonzalez-Rodriguez P, Annese V, Lopez-Barneo J & Pardal R. (2018). Fast neurogenesis from carotid body quiescent neuroblasts accelerates adaptation to hypoxia. *EMBO Rep* **19**.
- Peng YJ, Nanduri J, Wang N, Su X, Hildreth M & Prabhakar NR. (2025). Signal Transduction Pathway Mediating Carotid Body Dependent Sympathetic Activation and Hypertension by Chronic Intermittent Hypoxia. *Function (Oxf)* **6**.

Finally, including a graphical abstract would enrich the presentation of results.

EDITOR COMMENTS

Reviewing Editor:

Thank you for submitting your manuscript to The Journal of Physiology. The paper provides evidence of a novel mechanism contributing to carotid body plasticity evoked by exposure to chronic sustained hypoxia. The article is well written and very well presented with convincing evidence and limitations of the approaches used openly acknowledged.

Two experts have reviewed the manuscript and both are enthusiastic in their general overview of the study recognising the significance of the findings and potential relevance for the wider community interested in carotid body overactivity in disease states. Both referees have provided critiques, which should be very straightforward to address.

We thank the Reviewing Editor for their summary and very positive comments regarding our manuscript. We appreciate the opportunity to make changes to improve the manuscript.

Please note that R1 suggests moving a figure to supplemental files, but The Journal does not support supplementary material so these data can be presented as a Figure in the main manuscript. The significance of the data should be clarified to the referee. You included a supplementary Figure relating to the machine learning approach used for image analysis. This Figure can be incorporated into the main manuscript or you can position it in an appendix at the end of the article, but it will remain within a single document and will be published as part of the main manuscript.

Thank you for this helpful guidance on these matters. In terms of the machine learning immunohistochemistry figure, we have now decided to include this in the main manuscript. Although primarily a methods figure, this is a novel approach, and we think it will be used by others within the field to aid with analysis of similar datasets. Full descriptions are also provided in the methods to allow for implementation.

Regarding the second point. The *in vivo* figure in question (now Figure 8) demonstrates the effect of pharmacological CD73 inhibition on minute ventilation, mean arterial blood pressure (mABP) and heart rate before and after acute carotid sinus nerve section. The data is important for two key reasons. First, it demonstrates that inhibiting CD73 *in vivo* has physiologically relevant actions in CH animals and in particular decreases mABP. It builds on our *ex vivo* data and suggests that inhibition of CD73 activity in the carotid body can lead to important physiological consequences in the whole animal, hopefully aiding with future translation. Second, to establish that these *in vivo* observations caused by AOPCP are indeed carotid body dependent, the effect of the drug was investigated before and after acute carotid sinus nerve section. This type of acute nerve section cannot be performed in awake animals and requires anaesthesia. Evaluating the response soon after the nerve section also prevents any adaptation that may have occurred if the response

had been evaluated weeks after the surgery, following recovery. We think that this was a robust approach to show that some of the key physiological observations that we saw with CD73 inhibition were mediated by a decrease in carotid body function, thereby providing important mechanistic information. We do acknowledge limitations of using anaesthesia in the discussion, and have expanded upon this, but are confident that carotid body reflexes are preserved under these experimental conditions.

Both referees raise concern with sample sizes for some data sets and this point should be carefully addressed.

Thank you, we have responded to this point directly in comments to the senior editor and referees. Please see below.

Senior Editor:

Thank you for the submission of your research article to the Journal of Physiology for consideration. The article has been reviewed by experts in the field and found to be potentially acceptable for publication pending adequate revision to address all of the concerns raised. Please address all comments from the external referees and reviewing editor, as well as address the list of requirements for publication in the journal, including the statistical requirements, as outlined in this letter, the Instructions to Authors online, and the Journal's policy on rigour and reproducibility, found at the link below.

We thank the Senior Editor for this positive initial evaluation of our work and for the opportunity to make amendments to improve our manuscript.

The small sample size in some experiments (e.g., Figures 1 and 2) raises concern about adequate statistical analysis to support conclusions based on these data. It is recommended to increase the sample size to meet an adequate power of the analysis as needed.

1. Include in the Ethical Approval section a statement that the investigators understand the ethical principles under which the journal operates and that their work complies with the animal ethics checklist as outlined by the journal.

Thank you. We have added a statement to the Ethical approval section (p6) as follows-

“All investigators understand the Journal's policies regarding animal experiments and adherence to these policies was maintained throughout this research. All work complies with the animal ethics checklist as outlined by the Journal.”

2. Sample size must avoid pseudo-replication. (using multiple measurements from the same subject as if they were independent replicates, leading to an inflated sample size and potentially misleading statistical results). Clarify how the statistical sample size was

determined for each statistical test. Use mixed-effects models or hierarchical models when you have repeated measurements from the same experimental unit.

Thank you for this comment. In table 1, for all gravimetry, haematocrit, haemoglobin and growth rate data, a single experimental subject was considered as a single animal. Similarly, for all *in vivo* physiological measurements of breathing, mABP and heart rate in Figures 2, 7, 8, 9 and table 2, a single experimental subject was considered a single animal. For IHC experiments, multiple carotid body sections were analysed from a single animal (Figures 2 and 3). We have now made sure that exact numbers of sections and numbers of animals are stated in the figure legends for each experiment in accordance with The Journal's statistics policy. Furthermore, for nerve experiments sometimes multiple single fibres were recorded from the same animal (Figures 2, 4, 5 and 6). Specific numbers of fibres and numbers of animals are stated in the figure legends. As suggested by the Senior Editor, to account for this, we have now performed additional nested t-tests or nested one-way ANOVAs on data presented in the following Figures: Fig. 2B, C and E; Fig.3B, C and D; Fig.4. H and I; Fig. 5 D and E; Fig. 6G. Where necessary, p values have been amended in these figures and figure legends have been changed as appropriate. We think that this approach is the most appropriate as it allows us to show all data for every section/fibre (aiding with transparency, and showcasing variability), but this hierarchical statistical analysis now takes into consideration that some measurements are related and not totally independent. The use of nested t-tests or nested one-way ANOVAs has been added to the Data Analysis section of the methods (p14) and reads as follows-

“Where multiple carotid body tissue section or nerve fibre recording measurements were taken from the same experimental animal a nested t-test or nested one-way ANOVAs was used to take into consideration that some measurements were related and not totally independent.”

Regarding statistics, we have now also amended Table 2 (p22) so that all exact p values are clearly stated rather than using symbols to denote significance.

3. The small sample size in some experiments (e.g., Figures 1 and 2) raises concern about adequate statistical analysis to support conclusions based on these data. It is recommended to increase the sample size to meet an adequate power (beta) of the analysis as needed.

Additional IHC experiments on carotid body tissue taken from N and CH animals for CD73/CD31 protein expression analysis requires the acquisition of a new PPL from the UK Home Office. Whilst the application for a new PPL is currently underway, it could take 6-12 months to secure. Furthermore, the investigator that had expertise in performing these experiments has now completed their PhD and has joined a new research group. Therefore, to perform these experiments would require further funding acquisition and training, which again is likely to take time. Therefore, we will not be able to perform these extra experiments in a timely manner. We do think that although the IHC data is from a small number of animals there is still high-quality evidence to support the conclusion that the number of CD73⁺TH⁺ cells in the carotid body increases in CH. The idea that CD73 is

functionally important in the newly matured TH⁺ cells following CH is strongly backed by both the *ex vivo* carotid sinus nerve data and the *in vivo* physiological measurements. Thus, we are confident in our conclusion that CD73 plays an important functional role in the CH carotid body and think that there is sufficient data to support this. That said, we do appreciate that the future validation of the IHC CD73 would still be useful and therefore we have added the following section into the discussion (p25) as follows-

“However, our data does indicate that in the CH adapted carotid body, the newly formed TH positive type I cells co-express CD73. We acknowledge that CD73 and TH protein localisation data was analysed from a small number of animals and therefore future validation will be useful. Furthermore, quantification of CD73 protein expression in the different subtypes of cells within the carotid body using western blotting could be an important avenue to explore in the future.”

REFEREE COMMENTS

Referee #1:

Nathanael et al. evaluated whether the ecto-5-nucleotidase (CD73) participates in the carotid body hyperactivity in rats exposed to chronic sustained hypoxia (CH) for 10 days. They reported that CH increases the proportion of type I cells containing CD73 in the carotid body. Pharmacological targeting of CD73 and transmembrane adenylyl cyclases reduced basal carotid body chemosensory hyperactivity in CH and normalizes the enhanced responses to acute hypoxia in an *in vitro* CB preparation. Inhibition of CD73 in CH-rats induced a carotid body related decrease in arterial blood pressure in normoxia, without abolishing cardio-respiratory responses to acute severe hypoxia. Thus, authors concluded that CD73 is important in promoting carotid body hyperactivity in CH animals, suggesting that selective targeting of CD73 and cAMP signalling may be useful to ameliorate the carotid body hyperactivity and improve cardiovascular outcomes in chronically hypoxic patients.

The work is original, and the Ms. is clearly written. The study design and robustness of the experimental data is appropriate. The conclusion is supported by several experiments using different techniques. This interesting work is a contribution to the area of research. However, there are some issues that authors need to address to improve Ms. and make it suitable for publication.

We thank the referee for their positive and enthusiastic assessment of our work, and for their insightful comments provided. We hope that our responses and the changes we have made to the manuscript sufficiently address all the points raised.

Major:

Fig 2. It seems that CH has a small impact on CD7 cells in the carotid body. Figure 2A and 2B showed that there were no significant changes in TH cells or the percentage of CD73-positive cells in the CB. However, the overlap between CD73-positive cells and TH cells yielded a p-value of 0.0049. This result may reflect the limited sample size (n=3) used in the analysis. Why not measure CD73 using qPCR?

We thank the referee for this question. We opted to use an immunohistochemical approach as opposed to using qPCR for the following reason. The work by Salman and colleagues have already demonstrated that exposure to chronic hypoxia increases CD73 mRNA in whole carotid body extracts. Whilst qPCR does provide valuable quantitative information regarding gene expression, it does not offer any spatial information regarding CD73 protein localisation. Here, the use of immunohistochemical detection of CD73 protein on intact adult carotid body sections allowed us to probe alterations in cellular distribution of CD73 and TH caused by chronic hypoxia. We think that our data does provide evidence that in response to chronic hypoxia, new TH+ positive cells co-express CD73, thereby increasing the overall population of CD73+TH+ cells in the carotid body. The functional importance of this is then showcased by the findings that CD73 inhibition has a remarkable impact on reducing carotid sinus nerve activity. That said, we do agree that this IHC method is not the best to quantify protein level and that in future, selective quantification of CD73 protein using western blotting from the different cell types within the carotid body should be undertaken.

In view of these points raised we have added the following text to the discussion (p24-25)-

“It has been previously shown that whole carotid body CD73 mRNA expression is elevated by CH (Salman et al., 2017). The current study used immunohistochemical techniques to explore changes in the spatial arrangement of CD73 protein and/or modifications in cellular distribution of CD73 in the adult carotid body, induced by CH.”

Regarding the sample size, we acknowledge that the data is from a relatively small number of animals but think that it does provide evidence that CH increases the proportion of CD73+TH+cells. However, we have now added the following section into the discussion (p25)-

“However, our data does indicate that in the CH adapted carotid body, the newly formed TH positive type I cells co-express CD73. We acknowledge that CD73 and TH protein localisation data was analysed from a small number of animals and therefore, future validation will be useful. Furthermore, quantification of CD73 protein expression in the different subtypes of cells within the carotid body using western blotting could be an important avenue to explore in the future.”

Fig 7. Why were the effects of AOPCP before and after CBNX evaluated in anesthetized rats? (rats were initially anaesthetized with 2-5% isoflurane in O₂, and anesthesia was maintained with Alfaxalone). It is well known that anesthesia impairs sympathetic mediated

responses including the vascular response to hypoxia. In freely moving rats, hypoxia increases the arterial blood pressure. This is a serious limitation because an important central sympathetic depression will have occurred. The value of these experiments is unclear. Consider presenting the results in a supplemental figure.

The *in vivo* figure in question (now Figure 8) demonstrates the effect of pharmacological CD73 inhibition on minute ventilation, mean arterial blood pressure (mABP) and heart rate before and after acute carotid sinus nerve section. The data is important for two key reasons. First, it demonstrates that inhibiting CD73 *in vivo* has physiologically relevant actions in CH animals and in particular decreases mABP. It builds on our *ex vivo* data and suggests that inhibition of CD73 activity in the carotid body can lead to important physiological consequences in the whole animal, hopefully aiding with future translation. Second, to establish that these *in vivo* observations with AOPCP are indeed carotid body dependent, the effect of the drug was investigated before and after acute carotid sinus nerve section. This type of acute nerve section cannot be performed in awake animals and requires anaesthesia. Evaluating the response immediately after the nerve section also prevents any adaptation that may have occurred if the response had been evaluated days/weeks after the surgery. We think that this was a robust approach to show that some of the key physiological observations that we saw with CD73 inhibition were mediated by a decrease in carotid body function, thereby providing important mechanistic information. Therefore, we would prefer to keep this figure in the article. However, we have now expanded upon this as a limitation in the discussion (p30):

“The in vivo cardiovascular-respiratory experiments were performed on anaesthetised rats using Alfaxalone as the maintenance intravenous anaesthetic. This anaesthetic was used as it has been previously shown to preserve carotid body mediated cardiovascular-respiratory reflexes initiated by hypoxia, hypercapnia and hypoglycaemia (Thompson et al., 2016; Holmes et al., 2018). It also allowed us to assess the impact of CD73 inhibition on cardiovascular-respiratory function before and immediately after CSNX, preventing any adaptation and providing valuable mechanistic information. Whilst the use of anaesthesia is necessary for such invasive surgical procedures, a degree of central depression will have occurred, leading to some attenuation of sympathetic mediated vascular responses to hypoxia. In addition, the HVR’s measured in both groups appear to be slightly smaller than those measured in our awake unanaesthetised experiments using WBP. Thus, the effects of CD73 inhibition on cardiovascular and respiratory function in normoxia and hypoxia that we report here may not absolutely reflect what would occur in an awake and freely moving animal.”

What mechanisms led to reduced arterial pressure by AOPCP in N and CH animals? What is the implication that AOPCP did not abolish cardiovascular-respiratory reflex responses to severe hypoxia in both N and CH rats? Is it possible that there are distinct mechanisms involving CD73 and adenosine effects depending on the severity of hypoxia?

Thank you, this is a key point. We think that in normoxia, the decrease in blood pressure caused by AOPCP is due to inhibition of basal carotid body nerve activity. This was based

on the finding that the fall in mABP caused by AOPCP was attenuated following CSNX. This was also apparent in N animals. However, the fall in mABP caused by AOPCP was not completely abolished after CSNX. Potential additional mechanisms have been suggested in the discussion (p25) (see section below). In CH animals AOPCP also induced a small decrease in breathing in normoxia, which was not observed after CSNX. In our *ex vivo* nerve data, although basal carotid body activity is strongly attenuated by AOPCP and SQ22536, there is still a response to hypoxia, albeit left shifted. Indeed, if the hypoxic stimulus is severe enough the nerve frequency, can still rise to a maximal level. This is consistent with our observations that *in vivo* cardiovascular-respiratory reflexes were largely preserved in the presence of AOPCP when exposed to the severe level of hypoxia used here (FiO₂ 8%). In view of the left shift of the hypoxic response curve caused by AOPCP and SQ22536, we do agree with the referee that CD73, adenosine and tmAC activity is more involved in setting basal chemoafferent activity and the early mild-moderate phases of the hypoxic response rather than severe hypoxia. This is also in agreement with the work by Conde and Colleagues who we have now referenced. We have added the following sections to the discussion (p27 and p29) to cover these points raised-

*“Although basal carotid body activity in CH was attenuated by AOPCP, chemoafferent frequency was still able to increase during hypoxia, albeit the response being left shifted. Indeed, if the hypoxic stimulus is severe enough the nerve frequency can still rise to a maximal level. This is consistent with our *in vivo* observations that cardiovascular-respiratory reflexes were largely preserved in the presence of AOPCP when exposed to a single level of severe hypoxia (F_iO₂ 8%). As well as providing important safety information, this data suggests that in the CH carotid body, CD73 and adenosine are more involved in setting basal chemoafferent activity and the mild-moderate phases of the hypoxic response. The importance of adenosine in regulating the early phase response to hypoxia in N animals has been reported previously (Conde et al., 2012).”*

*“In our model of CH, we showed that acute *in vivo* pharmacological inhibition of CD73 in normoxia successfully decreased arterial blood pressure acting predominantly via the carotid body. This was based on the finding that the fall in mABP caused by AOPCP was attenuated following CSNX. This was also apparent in N animals. However, in both N and CH the decrease in mABP by AOPCP was not completely abolished following CSNX, suggestive of additional, albeit more minor, mechanisms. We cannot rule out potential direct actions of AOPCP on the vasculature. Since AOPCP can cross the blood-brain barrier, carotid body independent effects may involve modulation of adenosine pathways within the medulla, such as the NTS and RVLM, regions known to integrate sympathetic output (Thomas & Spyer, 1996; Dale et al., 2002).”*

Minor:

- 49 carotid body hyperactivity associated with CH. Therefore, the current project is investigated.

Work or study seems more appropriate that project.

Thank you, we have changed this to “work” as suggested in the abstract (p3)

- 55 within the carotid body. Pharmacological inhibition of CD73 using AOPCP completely. Please define AOPCP, α , β -methylene ADP.

Thank you, we have now done this as requested (p3).

- 56 carotid body chemoafferent hyperexcitability, augmented hypoxic sensitivity and abolished CH-induced basal chemoafferent hyperactivity.

Please, define hyperexcitability and hyperactivity. They are sometimes used interchangeably.

Thank you. Both really mean the same thing- an unusually high level of chemoafferent firing frequency. To avoid confusion, we have now replaced “hyperexcitability” with “hyperactivity” throughout. This should help with clarity.

- 58-59 adenylyl cyclase (tmAC) inhibitor SQ22536. CD73 antagonism in vivo reduced arterial blood pressure in a carotid body and dose-dependent manner in normoxia, What does mean "in a carotid body and dose-dependent manner in normoxia?"

For clarity we have changed this sentence in the abstract (p3) to read as follows-

“In vivo administration of AOPCP to CH animals caused a dose-dependent decrease in arterial blood pressure, that was attenuated following carotid sinus nerve section (CSNX)”.

Similarly, we have amended key point 4 (p2) so that it now reads-

- *“In vivo inhibition of CD73 induces a dose-dependent decrease in arterial blood pressure in normoxia, that is attenuated following carotid sinus nerve section. However, AOPCP does not abolish cardiovascular-respiratory responses to severe hypoxia*

We have also had to slightly amend key point 5 (p2) to keep within the 150 word limit.

- *“CD73 and tmAC are important in promoting carotid body hyperactivity in CH. Selective targeting of CD73 and cAMP signalling may safely reduce carotid body hyperactivity and thus improve cardiovascular outcomes in chronically hypoxic patients”*

- 69-73 Carotid body hyperactivity is recognised as a key factor in the development and progression of cardiovascular pathology in multiple illnesses including obstructive sleep apnoea, chronic heart failure, essential hypertension, diabetes, obesity, chronic inflammation and COPD (Incognito et al.; Schultz et al., 2015; Phillips et al., 2018; Shin et al., 2019; Conde et al., 2023; Felipe et al., 2023; Prabhakar et al., 2023).

Authors cited most relevant works; however, some important contributions are missed (i.e.,

Abdala AP. J Physiol, 590 (17), 4269-4277. doi: 10.1113/jphysiol.2012.237800, Zera et al. J Physiol. 2025 603(9):2487-2516. doi: 10.1113/JP285081. Iturriaga R. J Physiol. 2023 601(24):5495-5507. doi: 10.1113/JP284112).

Thank you, we have now added these 3 additional references to this section in the introduction (p4) as suggested.

- 254 -257 superfused with the same physiological salt solution mentioned above. Basal chemoafferent activity was measured in a PO₂ of 300mmHg and PCO₂ of 40mmHg. This PO₂ was considered normoxic as this level has been shown to generate basal carotid body discharge of 0.25-1.50Hz (Holmes et al., 2014).

What is the rationale to superfuse the CB with a saline equilibrated at PO₂ of 300 mmHg? The rat CB *in vitro* superfused preparation work at lower PO₂ levels in the superfusate. Why do authors consider that high PO₂ as normoxia? Why is the frequency of discharge of 0.25 -1.5 Hz considered normoxia? Is it likely that the measured frequency depends on the window used to select action potential?

Thank you for raising this point. All reported nerve data are from isolated single fibres. Today's software enables sophisticated spike discrimination from recordings of few-fibre preparations, to allow for isolation of data from a single spike. We prefer to use single fibre recordings as these enable a greater understanding of potential underlying mechanisms of chemotransduction that the, more commonly utilised, multi-fibre preparations do not. This is because, with single fibre preparations, any increase or decrease in recorded discharge frequency can be attributed directly to changes in stimulus-response characteristics, whereas with a multi-fibre preparation there is always the underlying risk of fibre recruitment and/or fibre drop out affecting the discharge rate. Importantly, most basal single fibre chemoafferent activity is reported to be in the range of approximately 0.25-1.5 Hz (although there are some outliers with higher frequencies) in the rat carotid body when measured *in vivo* in arterial normoxia by Vidruk and colleagues (Vidruk *et al.*, 2001). *et al.*, 2001). In our *ex vivo*, superfused preparations there is a need to establish a greater PO₂ diffusion gradient than would be required for a perfused preparation, such that when the superfusate PO₂ is set at 300mmHg, the great majority of basal single fibre frequency lies within the *in vivo* normoxic range of 0.25-1.5 Hz range (e.g. see N group in Fig. 2E). Thus, we consider a PO₂ of 300 mmHg a normoxic PO₂ for this preparation although we accept there will be some variation across the tissue. We have amended a section in the methods (p9) that now reads as follows-

*“Basal chemoafferent activity was measured in a PO₂ of 300mmHg and PCO₂ of 40mmHg. PO₂ was considered normoxic for an *in vitro* superfused preparation, as this level has been shown to generate a basal single-fibre chemoafferent discharge frequency within the range of 0.25-1.50Hz (Holmes *et al.*, 2014) in this *in vitro*, superfused preparation. This is consistent with the observed *in vivo* single-fibre chemoafferent frequency range recorded in healthy rats during arterial normoxia (Vidruk *et al.*, 2001).”*

- 413- 418 During graded hypoxia chemoafferent frequency was augmented in CH and the PO₂-response curve for CH was right-shifted (Fig. 1F and G). This shift was quantified by measuring the superfusate PO₂-required to achieve a discharge frequency of 5 Hz, which was higher in the CH group (N: 65{plus minus}16 mmHg vs CH: 100{plus minus} 39 mmHg, p=0.0032, unpaired Student's t-test). Thus, as well as having basal hyperactivity, CH carotid bodies displayed an increase in hypoxic sensitivity.

How do you define sensitivity and reactivity?

Throughout the manuscript we have tried to distinguish between two concepts-

Hyperactivity- referring to an unusually high level of single fibre chemoafferent frequency under normal (i.e. normoxic) conditions

Increased sensitivity- a larger or earlier onset of a response when exposed to a specific stimulus, in this case hypoxia.

We have carefully checked through and made minor adjustments to make this distinction clearer throughout the manuscript

Fig 3, It seems that the frequency of chemosensory discharges is higher than the one measured (1 Hz in normoxia and 2 in CH-CBs? Why in Fig 3F, the size of the action potential was different?

In this panel (Now Fig.4A), the frequency histograms presented (lower) correspond to an isolated single fibre from the raw recording (upper). It is true that there could be multiple fibres contributing to the raw recording (upper) but data is extracted and analysed specifically from a single isolated fibre. The single fibre discrimination is demonstrated by the overlay of signals displayed inset. We have checked and the frequency histograms presented in Fig 4A are correct for both N and CH.

Regarding the second point, in what is now Fig. 4F, as this is an extracellular recording, the signal amplitude is not a real reflection of the size of action potential amplitude. These extracellular signals are in the μ V range and vary greatly between recordings depending on the position/orientation of the recording electrode relative to the propagating action potential waveform along the neurone, the amount of surrounding connective tissue and the stability of the applied suction. To accurately measure changes in action potential amplitude (that would be in the mV range) would require intracellular recordings from the cell bodies in the petrosal ganglion, which was not the focus of the current work.

In Fig 3B and C, at what PO₂ the effects of CH and AOPCP on the frequency of discharge was significant? Is this important for determining changes in sensitivity?

Thank you for raising this. We think that the referee is referring to the original Fig. 4B and C, which is now Fig. 5B and C, apologies if this is incorrect. The statistics presented give the overall difference between the 3 curves, suggesting that the overall chemoafferent

response to hypoxia is elevated in CH and then restored in the presence of AOPCP, which we think is the key message of this part of the figure. We think that adding all individual p values at all the different PO₂s comparing the 3 curves is not needed to make this point and could potentially confuse the reader. However, we agree that it is important to consider some sort of measure of O₂ sensitivity. When referring to sensitivity we think this is best evaluated by looking at the PO₂ onset of the response to hypoxia. For this we choose to compare the PO₂ required to achieve a single fibre frequency of 5Hz as this is well above the basal frequency for both N and CH and lies on the early part of the exponential phase of the hypoxic response curve, corresponding to the onset of the hypoxic response. This data is presented in Fig.5D and E and statistical comparisons are made between the three groups. We think that this more clearly demonstrates changes in O₂ sensitivity rather than plotting the change in frequency per mmHg decrease in O₂, which could have been an alternative approach. Furthermore, it isn't possible to plot the exact P50 for these curves as we intentionally don't achieve the maximum frequency response for each fibre to avoid potential preparation run-down.

In Fig 1D the frequency is expressed in Hz in N, but in CH is expressed in μ V?

Thank you, we agree that this could be explained more clearly in the figure legend. This is now Fig. 2. In Fig2D, raw discharge is shown in the upper panels for both N and CH, the axis scale (on the left) is the same for N and CH and the unit is μ V. Frequency is shown in the lower panels for both N and CH, the axis scale on the left is the same and the unit is Hz. We have altered the figure legend 2 (p41) to hopefully make this clearer. It now reads as follows- *“Example traces of basal chemoafferent activity from N (left) and CH (right). Raw discharge (μ V) is shown in the upper panels and frequency histograms (Hz) in the lower panels. Axes scales are the same for N and CH.”*

We also think that panel 2F could be slightly clearer. Therefore, in Fig. 2F we have added a raw discharge scale on the right-hand side to the upper panel. The axis scale is the same for N and CH. We have amended the figure legend 2 (p41) as needed which now reads as follows- *“Example recordings of chemoafferent responses to graded hypoxia for N (left) and CH (right). Raw discharge (μ V) is shown in the upper panels (axis on the right) and the superfusate PO₂ (mmHg) is overlaid in red (axis on the left). Frequency histograms (Hz) are shown in the lower panels (axis on the left). All axes scales are the same for N and CH.”*

816-821 Pharmacological modulation of the carotid body: A safer alternative to CSNX or ablation? In animal models of carotid body hyperactivity, CSNX has been shown to reduce excessive sympathetic outflow, lower blood pressure, and in hypercaloric type 2 diabetic rats, restore insulin sensitivity (Abdala et al., 2012; Ribeiro et al., 2013; Sacramento et al., 2017).

In chronic intermittent hypoxia models of obstructive sleep apnea, CB ablation and CSNX reduce hypertension and autonomic changes (Pereyra K. Am J Physiol Lung Cell Mol

Physiol. 2025;328(3): L357-L371; Del Rio R. Hypertension. 2016;68(2):436-45).

Thank you we have now added a sentence alluding to this and added both suggested references to the discussion (p28).

Referee #2:

Upon activation, the carotid body (CB) has been proven to trigger potent respiratory reflexes as well as being one of the most potent stimulants of cardiovascular sympathetic activity. In this remarkable article of integrative physiology, the authors conduct an exhaustive and elegant study focusing on chronic hypoxia (CH)-induced CB hyperactivity, its systemic consequences, and how they can be reversed pharmacologically. They provide an in-depth, comprehensive physiological perspective of the chemoreception field.

In particular, the authors explore the role of CD73, one of the ectonucleotidases expressed in the CB. They demonstrate that basal chemoafferent CB hyperactivity in CH can be fully reversed by the pharmacological inhibition of CD73 using AOPCP. Furthermore, AOPCP normalises the enhanced hypoxic sensitivity of the CB exposed to CH, which is responsible for an exaggerated hypoxic ventilatory response (HVR). Unlike the effects of previous CB treatment described in the literature, this study reveals that, while pharmacological inhibition of CD73 within the CB does reverse maladaptive chemoafferent activity in normoxia, it does not abolish cardiovascular-respiratory reflex responses to severe hypoxia.

This effect seems to be linked to the activity of adenylyl cyclase (AC), which produces an excess of cAMP - the substrate for CD73 - because the inhibition of AC in parallel experiments using SQ-22536 reproduces the effects of AOPCP. The expression of CD73 by CB cells and its increase in type I cells during chronic hypoxia (CH) are well-known facts, as is the role of cAMP and adenylyl cyclase in the CB response to hypoxia, which can be blocked by SQ 22536. [Salman S, Vollmer C, McClelland GB, Nurse CA. Characterization of ectonucleotidase expression in the rat carotid body: regulation by chronic hypoxia. *Am J Physiol Cell Physiol.* 2017 Sep 1;313(3):C274-C284]. And, in my opinion, the authors do not make it sufficiently clear in the discussion.

However, the most important finding is that the authors unveil a novel function of CD73 in promoting CB chemoafferent hyperactivity in CH. Their results provide evidence that the increased basal chemoafferent frequency observed in CH is highly dependent on CD73 and can be completely reversed by AOPCP. As there is no evident enhanced sensitivity to adenosine, the authors hypothesise that a greater pool of extracellular adenosine exists in CH due to the overexpression of CD73. This reinforces the role of ATP/ ADP/ adenosine pool in activating the CB tripartite synapse, which includes afferent activation of the CSN and its plasticity, in CH, as described by Colin Nurse and cols.

The study is highly relevant because the CB is not only an oxygen sensor, but also

responds acutely to other stimuli besides oxygen, such as glucose. Feedback from blood-borne hormones that regulate metabolism and even ROS and inflammatory mediators also influence its response. Thus, the CB has been shown to be a major driver of the heightened sympathetic activity associated with several cardiometabolic diseases. Therefore, understanding the mechanisms of activation, cellular transduction and intercellular signalling within the CB is crucial for identifying novel drugs and interventions to reduce CB hyperactivity in disease.

Finally, despite the authors' acknowledgment of limitations in their COPD model, the value of this study remains intact, as preclinical models are instrumental in elucidating the physiology and pathophysiology of CB.

We thank the referee for their very thorough and positive assessment of our work, and for their helpful suggestions to improve the manuscript.

We agree that we could have done a better job at showcasing previous work evaluating the importance of CD73 in the carotid body in the discussion. In view of this we have added multiple sections and references to the discussion (p24-25) as follows-

“CD73 is a key enzyme involved in purine metabolism, specifically converting AMP into adenosine. It has been reported that higher levels of CD73 mRNA are expressed in the carotid body compared with other tissues including brain, super cervical ganglion and petrosal ganglion (Salman et al., 2017). CD73 protein is also expressed in the carotid body, specifically in type I and type II cells (Salman et al., 2017; Sacramento et al., 2019). CD73 functions to control extracellular adenosine concentration following release of ATP from type I and type II cells (Conde & Monteiro, 2004; Murali & Nurse, 2016; Leonard et al., 2018). In the normoxic unadapted carotid body, our preceding studies revealed that CD73 is important in establishing sensitivity to hypoxia and hypercapnia (Holmes et al., 2015; Holmes et al., 2018).”

“It has been previously shown that whole carotid body CD73 mRNA expression is elevated by CH (Salman et al., 2017). The current study used immunohistochemical techniques to explore changes in the spatial arrangement of CD73 protein and/or modifications in cellular distribution of CD73 in the adult carotid body induced by CH.”

MINOR COMMENTS

There is little to say about the 'Methods' section. The experiments are well designed and remarkably well described. However, some concerns have arisen.

- The authors describe the treatment of rats as chronic sustained hypoxia. Since intermittent hypoxia can also be applied chronically, renaming their experimental approach 'chronic sustained hypoxia' (CSH) clarifies this.

We appreciate this comment. However, we think that the use of CH is acceptable and commonly used within the field to describe this type of persistent exposure to a lower FiO₂. However, we have added/expanded the following sentences to the methods (p6) to help clarify the models and groups.

“The FiO₂ was then maintained at 12% for 9-10 days. These animals are referred to as being chronically hypoxic (CH), as they are exposed to chronic sustained hypoxia.”

“Control rats were kept in ambient air for the same duration and received the same enrichments. These animals are referred to as normoxic (N).”

Why is 10 days of hypoxia treatment used in this CH model?

Thank you. We agree that in the discussion we could provide more information/validation/rationale about our CH model. Therefore, we have now added the following paragraph to the discussion (p28-29)-

“Our model of CH involved exposing animals to an FiO₂ of 12% for 9-10 days. This is a relatively mild stimulus intensity compared to the more commonly used FiO₂ of 10%. However, an FiO₂ of 12% was selected to induce milder levels of hypoxia, more in line with those observed in COPD patients. The 9-10 day duration was based on previous studies showing that the increase in carotid body chemoafferent discharge frequency peaks at approximately 9 days of CH and does not rise further (Chen et al., 2002). Importantly, our model successfully induced key cardiovascular adaptations seen in COPD patients including an elevation in haematocrit, haemoglobin and right ventricular hypertrophy. Furthermore, it led to a robust rise in basal carotid body chemoafferent activity, an augmentation in carotid body hypoxic sensitivity and a rise in baseline minute ventilation. Therefore, we think that this is a valid model to assess cardiorespiratory and carotid body adaptation to mild levels of chronic sustained hypoxia, possibly more relevant to COPD patients.”

- Another question concerns the total number of animals used in the study and whether they were used for several non-interfering measures.

The total animal number used in the study was 84. We have stated in the methods for clarity (p6). An animal was generally only used for one experimental type- IHC, chemoafferent recordings or in vivo cardiovascular-respiratory reflexes. All animals were terminated using a schedule 1 method. Chemoafferent recordings were not taken from animals who had already been through in vivo cardiovascular-respiratory reflex studies where the animal had already been administered AOPCP. These were separate animals. Furthermore, a single carotid body was only exposed to either AOPCP or SQ22536 or adenosine to eliminate lasting drug effects/interactions, impacting on the results. However, to reduce animal numbers both carotid bodies from a single animal were used for nerve experiments. In these cases, one carotid body was used for an AOPCP experiment and the other from the same animal for SQ22536 or adenosine experiments. Furthermore,

sometimes the same carotid body was exposed to multiple varying concentrations of AOPCP (e.g in Fig. 4 and 5). Where this was done, a complete drug washout or final hypoxic response was performed to show that effects of the drug were fully reversible (please see Fig. 4A and Fig. 5A). Cardiac tissue was taken following confirmation of death and not from those animals who had already undergone in vivo drug experiments.

We have added the following sentence to the methods (p11) to help clarify-

“To reduce animal numbers, both carotid bodies were used from a single animal for chemoafferent recordings. However, a single carotid body was only exposed to one of the drug agents, either AOPCP, SQ22536 or adenosine.”

Some animals who had undergone awake breathing experiments using whole body plethysmography also went on to have carotid bodies isolated for measurements of chemoafferent activity or IHC, 1 day later. If this was the case animals were placed back into their home chamber and exposed to FiO₂ 12% for CH, or air for N. The WBP procedure takes approximately 1-2 hours and we do not think that this would be sufficient to cause any reversal of the adaptation to CH.

We have added the following section to the whole body plethysmography methods to help with clarity (p11)-

“Following experimentation, animals were either culled by exposure to carbon dioxide gas in a rising concentration, death confirmed by cervical dislocation, or placed back in their home cage at 12% F_iO₂ for CH and ambient air for N. In the latter instance, animals subsequently had carotid bodies isolated one day later for analysis of chemoafferent activity or CD73 protein using immunohistochemistry.”

Line 202 says: The relatively high pixel intensity of RBC AF and complexity of the images made regular thresholding and other conventional methods to remove this false-positive signal unsuitable. To address this, a machine learning approach was employed using the Python-based software, ilastik (Berg et al., 2019). For training, immunofluorescence images were manually annotated to label RBC pixels. This process was repeated across approximately four images to train the model. Once trained, the software predicted pixel classifications and batch-processed subsequent images.

Given my lack of knowledge in machine learning, I have a question: Is it enough to use four images to train the model?

We thank the referee for this question. The basis of this machine learning approach is defined in more detail in Berg et al 2019, which is referenced in the main manuscript. In theory, the amount of images used to train the model is entirely dependent on whether the software is able to correctly detect what you set out for it to detect. This would depend on the variation of the staining between your samples. The more variation, then perhaps the software needs more images to effectively be trained. In our case, the staining was

consistent between samples and therefore we utilised 1 image across our 3 biological replicates plus an additional image, which was sufficient for the software to accurately identify red blood cells across the rest of our data.

Line 408 says: Figure 1, CH induced an elevation in TH positive staining in the carotid body, but didn't appear to cause major alterations in carotid body size. There was also no evidence of significant vascular expansion since quantification of immunoreactivity for the vascular endothelial marker CD31 showed no difference between N and CH.

As the data are based on a small number of animals (N=3), it is difficult to ascertain whether there have been any changes in size or vascularisation, which are two hallmarks of CB plasticity in chronic hypoxia.

This could be the case. However, it is important to note that many other models of chronic hypoxia which do show changes in size and vascularisation, are based on more severe hypoxic intensities such as using an FiO₂ of 10%. Other models which have used an FiO₂ of 12% and show an increase in in carotid body volume and vascularisation have maintained this hypoxic level for 32 days (Clarke et al 2000). Therefore, this indicates that changes in carotid body function occur earlier than an elevation in size and vascularisation. As mentioned above we have now added a section to the discussion (p28-29) regarding our CH model and its validation.

Table I show normalization of LV and RV mass to body mass. However, to normalize LV and RV mass, it would be more accurate to calculate the left or right ventricle-to-tibia length (LV/TL (mg/mm)) ratio because rat body weight changes after CH.

We agree that normalisation against tibial length is useful and many groups use this to determine RV/LV hypertrophy throughout the field. We are unfortunately unable to perform this analysis now as we don't have access to the tibias from these animals. However, we are still very confident that there is RV hypertrophy which is a key feature of CH adaptation. Even though CH animals are indeed lower in mass, the absolute mass of the RV is still significantly higher than normoxic controls (Table 1, p16). Furthermore, Fulton's index is another common measure used to indicate RV hypertrophy, which doesn't use animal mass. Again, the large elevation in Fulton's index in the CH group compared to N points towards significant RV hypertrophy (Table 1, p16).

Figure 7. Part 'B-D' in the figure does not correspond to the text in the legend.

Thank you very much for spotting this. We have amended this figure (now figure 8) so that the labels and panels in the figure now match with the descriptions in the figure legend (p44). Similar alterations have also been made to figures 7 and 9.

- There are four references that are incomplete:

- Bishop T & Ratcliffe PJ. (2025). HIF2 α : the interface between oxygen-sensing systems in 909 physiology and pathology. *Physiology*.
- Incognito AV, Barioni NO, Sharkey KA & Wilson RJA. Carotid Body Stress And Inflammatory Mediators Uniformly Excite Autonomic Output. *CircRes* 0.
- Sobrino V, Gonzalez-Rodriguez P, Annese V, Lopez-Barneo J & Pardal R. (2018). Fast neurogenesis from carotid body quiescent neuroblasts accelerates adaptation to hypoxia. *EMBO Rep* 19.
- Peng YJ, Nanduri J, Wang N, Su X, Hildreth M & Prabhakar NR. (2025). Signal Transduction Pathway Mediating Carotid Body Dependent Sympathetic Activation and Hypertension by Chronic Intermittent Hypoxia. *Function (Oxf)* 6.

Thank you for spotting this. These references are now complete.

Finally, including a graphical abstract would enrich the presentation of results.

Thank you. We agree that a graphical abstract would help to showcase the key findings of our study. We have now compiled this and attached as a separate file. The graphical abstract figure legend is also provided at the end of the manuscript.

References related to this response

- Chen J, He L, Dinger B, Stensaas L & Fidone S. (2002). Role of endothelin and endothelin A-type receptor in adaptation of the carotid body to chronic hypoxia. *Am J Physiol Lung Cell Mol Physiol* **282**, L1314-1323.
- Conde SV & Monteiro EC. (2004). Hypoxia induces adenosine release from the rat carotid body. *J Neurochem* **89**, 1148-1156.
- Conde SV, Monteiro EC, Rigual R, Obeso A & Gonzalez C. (2012). Hypoxic intensity: a determinant for the contribution of ATP and adenosine to the genesis of carotid body chemosensory activity. *J Appl Physiol* **112**, 2002-2010.
- Dale N, Gourine AV, Llaudet E, Bulmer D, Thomas T & Spyer KM. (2002). Rapid adenosine release in the nucleus tractus solitarii during defence response in rats: real-time measurement in vivo. *J Physiol* **544**, 149-160.
- Holmes AP, Nunes AR, Cann MJ & Kumar P. (2015). Ecto-5'-Nucleotidase, Adenosine and Transmembrane Adenylyl Cyclase Signalling Regulate Basal Carotid Body Chemoafferent Outflow and Establish the Sensitivity to Hypercapnia. In *Arterial Chemoreceptors in Physiology and Pathophysiology*, ed. Peers C, Kumar P, Wyatt

CN, Gauda E, Nurse CA & Prabhakar N, pp. 279-289. Springer-Verlag Berlin, Berlin.

Holmes AP, Ray CJ, Pearson SA, Coney AM & Kumar P. (2018). Ecto-5'-nucleotidase (CD73) regulates peripheral chemoreceptor activity and cardiorespiratory responses to hypoxia. *J Physiol-London* **596**, 3137-3148.

Holmes AP, Turner PJ, Carter P, Leadbeater W, Ray CJ, Hauton D, Buckler KJ & Kumar P. (2014). Glycogen metabolism protects against metabolic insult to preserve carotid body function during glucose deprivation. *J Physiol* **592**, 4493-4506.

Leonard EM, Salman S & Nurse CA. (2018). Sensory Processing and Integration at the Carotid Body Tripartite Synapse: Neurotransmitter Functions and Effects of Chronic Hypoxia. *Frontiers in physiology* **9**, 225.

Murali S & Nurse CA. (2016). Purinergic signalling mediates bidirectional crosstalk between chemoreceptor type1 and glial-like type2 cells of the rat carotid body. *J Physiol-London* **594**, 391-406.

Sacramento JF, Olea E, Ribeiro MJ, Prieto-Lloret J, Melo BF, Gonzalez C, Martins FO, Monteiro EC & Conde SV. (2019). Contribution of adenosine and ATP to the carotid body chemosensory activity in ageing. *The Journal of Physiology* **597**, 4991-5008.

Salman S, Vollmer C, McClelland GB & Nurse CA. (2017). Characterization of ectonucleotidase expression in the rat carotid body: regulation by chronic hypoxia. *Am J Physiol-Cell Physiol* **313**, C274-C284.

Thomas T & Spyer K. (1996). The role of adenosine receptors in the rostral ventrolateral medulla in the cardiovascular response to defence area stimulation in the rat. *Exp Physiol* **81**, 67-77.

Thompson EL, Ray CJ, Holmes AP, Pye RL, Wyatt CN, Coney AM & Kumar P. (2016). Adrenaline release evokes hyperpnoea and an increase in ventilatory CO₂ sensitivity during hypoglycaemia: a role for the carotid body. *The Journal of physiology* **594**, 4439-4452.

Vidruk EH, Olson EB, Jr., Ling L & Mitchell GS. (2001). Responses of single-unit carotid body chemoreceptors in adult rats. *J Physiol* **531**, 165-170.

Dear Dr Holmes,

Re: JP-RP-2025-289539R1 "Targeting ecto-5'-nucleotidase (CD73) and cAMP alleviates carotid body hyperactivity and reduces blood pressure in chronically hypoxic rats" by Demitris Nathanael, Andrew M Coney, Dhaifallah Saud Alotaibi, Prem Kumar, and Andrew P Holmes

We are pleased to tell you that your paper has been accepted for publication in The Journal of Physiology.

Yours sincerely,

Harold Schultz
Senior Editor
The Journal of Physiology

IMPORTANT POINTS TO NOTE FOLLOWING ACCEPTANCE OF YOUR PAPER:

- You can help your research get the attention it deserves! Check out Wiley's free Promotion Guide for best-practice recommendations for promoting your work at: www.wileyauthors.com/eoo/guide. You can learn more about Wiley Editing Services which offers professional video, design, and writing services to create shareable video abstracts, infographics, conference posters, lay summaries, and research news stories for your research at: www.wileyauthors.com/eoo/promotion.

- If you would like to receive our 'Research Roundup', a monthly newsletter highlighting the cutting-edge research published in The Physiological Society's family of journals (The Journal of Physiology, Experimental Physiology, Physiological Reports, The Journal of Nutritional Physiology and The Journal of Precision Medicine: Health and Disease), please click this link, fill in your name and email address and select 'Research Roundup': <https://www.physoc.org/journals-and-media/membernews>

EDITOR COMMENTS

Reviewing Editor:

Thank you for providing thorough responses to the original critiques and for your well-spirited, transparent and careful approach in revising the article to further improve it. All points have been satisfactorily addressed. Congratulations on an excellent study, providing mechanistic insight to carotid body sensitisation to chronic hypoxia, relevant to high altitude

adaptation and aberrant remodelling in the context of chronic cardiopulmonary diseases. We expect that the manuscript will be highly influential to the field.

Senior Editor:

The editors thank the authors for the final adjustments to the manuscript. The article is now accepted for publication. Congratulations on an interesting and insightful study. Please consider the Journal of Physiology for your future studies.

REFEREE COMMENTS

Referee #1:

This revised Ms is suitable for publication in J Physiol. Authors answered most of the comments.

Referee #2:

The authors have produced an exhaustive and excellent report, addressing all the points raised in the review and incorporating all the suggested corrections.

I have no further comments to add.